# Immune features are associated with response to neoadjuvant chemo-immunotherapy for muscle-invasive bladder cancer

Neoadjuvant cisplatin-based chemotherapy is standard of care for muscle-invasive bladder cancer (MIBC). Immune checkpoint inhibition (ICI) alone, and ICI in combination with chemotherapy, have demonstrated promising pathologic response (<pT2) in the neoadjuvant setting. In LCCC1520 (NCT02690558), a phase 2 single-arm trial of neoadjuvant chemo-immunotherapy (gemcitabine and cisplatin plus pembrolizumab; NAC-ICI) for MIBC, 22/39 patients responded (pathologic downstaging as primary outcome), as previously described. Here, we report post-hoc correlative analyses. Treatment was associated with changes in tumor mutational profile, immune gene signatures, and RNA subtype switching. Clinical response was associated with an increase in plasma IL-9 from pre-treatment to initiation of cycle 2 of therapy. Tumors harbored diverse predicted antigen landscapes that change across treatment and are associated with APOBEC, tobacco, and other etiologies. Higher pre-treatment tumor *PD-L1* and *TIGIT* RNA expression were associated with complete response. IL-8 signature and Stroma-rich subtype were associated with improved response to NAC-ICI versus neoadjuvant ICI (ABACUS trial, NCT02662309). Plasma IL-9 represents a potential predictive biomarker of NAC-ICI response, while tumor IL-8 signature and stroma-rich subtype represent potential predictive biomarkers of response benefit of NAC-ICI over neoadjuvant ICI. Future efforts must include additional independent biomarker discovery and validation, ultimately to improve the selection of patients for ICI-related treatments.

Bladder cancer is a common malignancy with over 80,000 new cases and 17,000 deaths annually in the United States[1]. Despite aggressive treatment including neoadjuvant chemotherapy and radical cystect-omy (RC), muscle-invasive bladder cancer (MIBC) is a potentially lethal disease with >50% of patients developing recurrent disease and cancer-related death. Immune checkpoint inhibitors (ICI) have been FDA approved for patients with advanced/metastatic disease; how-ever, only a subset of patients derive long-term benefit[2–5].

The incorporation of ICI in the neoadjuvant setting is a promising strategy to improve outcomes in MIBC with two recent trials of neoadjuvant ICI monotherapy reporting pathologic response rates (<pT2) of 55 and 66%, and pathologic complete responses (pCR) seen in 31 and 46%[6,7]. Adding ICI to first-line chemotherapy has been mostly unsuccessful in improving outcome for bladder cancer in the meta-static setting except for the recent report of the addition of nivolumab to gemcitabine-cisplatin[8]. However, we have observed high pathologic

e-mail: jonathan_serody@med.unc.edu; wykim@med.unc.edu; benjamin_vincent@med.unc.edu

response rates to combination chemo-ICI (NAC-ICI) in MIBC in LCCC1520 using gemcitabine and cisplatin plus pembrolizumab[9] and reported 14 of 39 (36%) patients experiencing pCR, and an additional 8 patients with pathologic downstaging, with improved recurrence-free survival seen in those patients with pathologic downstaging.

Prediction of response to neoadjuvant ICI and NAC-ICI is crucial since many patients do not benefit from treatment and significant toxicities are common. Previously identified features associated with ICI response include tumor mutation burden (TMB)[10], neoantigen burden[11,12], expression of antigen presentation machinery[13,14], DNA damage response (DDR) pathway mutations[15,16], PD-L1 expression[11,17], IFNγ signature[18], intratumoral heterogeneity[19,20], and microbiota features[21,22]. Models with genomic, molecular, and clinical variables can predict ICI response with moderate accuracy (AUC = 0.78–0.82) across multiple cancer types including bladder cancer[23,24]. However, there is not yet a comprehensive and accurate predictive model for chemo-ICI response, or a predictive model in the neoadjuvant setting. The interplay between chemotherapy and ICI has been studied, revealing that while chemotherapy can inhibit ICI by suppressing T cell function[25], chemotherapy can also assist ICI by increasing tumor inflammation[25] and inducing immunologic cell death[26].

In this work, we report molecular and cellular features associated with response and survival following neoadjuvant gemcitabine and cisplatin chemotherapy plus PD-1 inhibition with pembrolizumab in MIBC. Treatment is associated with changes in mutational profile, immune gene signatures, and RNA subtype switching of tumors. Clinical response is associated with an increase in plasma IL-9 from pre-treatment to initiation of cycle 2 of therapy. Tumors harbor diverse predicted antigen landscapes that change across treatment and are associated with APOBEC, tobacco, and other etiologies. Higher pre-treatment tumor PD-L1 and TIGIT RNA expression are associated with complete response. Finally, IL-8 signature and Stroma-rich subtype are associated with improved response rates to NAC-ICI compared to neoadjuvant ICI (from the ABACUS trial).

## Results

### Clinical features of NAC-ICI

A summary of the demographic, clinical, and pathologic findings in the LCCC1520 trial of pembrolizumab plus gemcitabine and cisplatin in MIBC has been previously published[9]. Out of 39 enrolled patients, 22 (56%) had a pathologic response (<pT2N0) and 14 (36%) had a complete pathologic response. From LCCC1520, we have multiple specimens (tumor, plasma, and PBMCs) and data types (DNA, RNA, flow cytometry, and plasma analytes) across three timepoints (pre-treatment/TURBT, cycle 2, and post-treatment/cystectomy) to evaluate potential biomarkers of response (Fig. 1A). Response (pathologic response) was not associated with pre-treatment clinical T stage (Fig. 1B) or baseline ECOG performance status (Fig. 1C), or the occurrence of an adverse treatment-related event that required cessation of chemotherapy (Fig. 1D), although no partial responders (pathologic response <pT2N0 without pCR) had an adverse event.

### Changes during NAC-ICI treatment

To understand how NAC-ICI alters the tumor microenvironment (TME), we analyzed changes in non-pCR tumors from pre- to post-treatment. We first analyzed changes in TMB, one of the better pan-

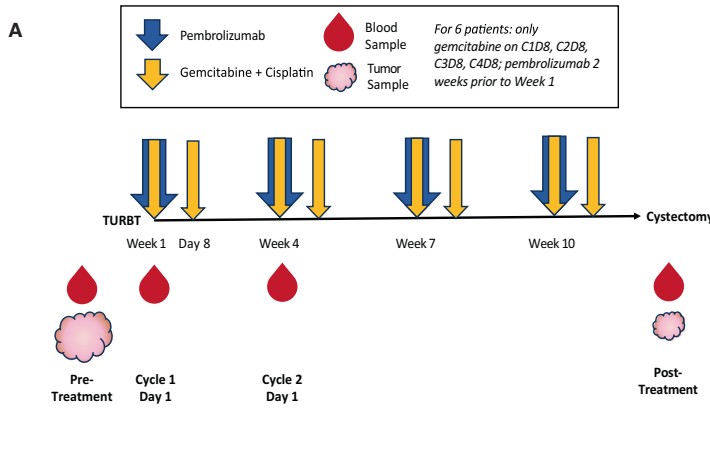

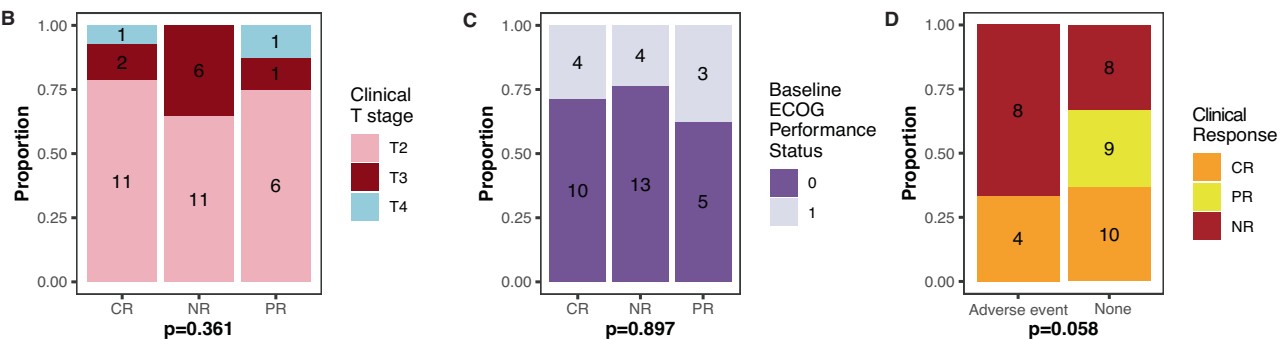

**Fig. 1 | LCCC1520 clinical data. A** Overview of LCCC1520 clinical trial timeline and data collection timepoints. **B** Clinical T stage and (**C**) baseline ECOG performance status by response (n = 39, Fisher's exact tests, no FDR correction). **D** Response by occurrence of an adverse treatment-related event that required cessation of chemotherapy (n = 39), Fisher test, no FDR correction.

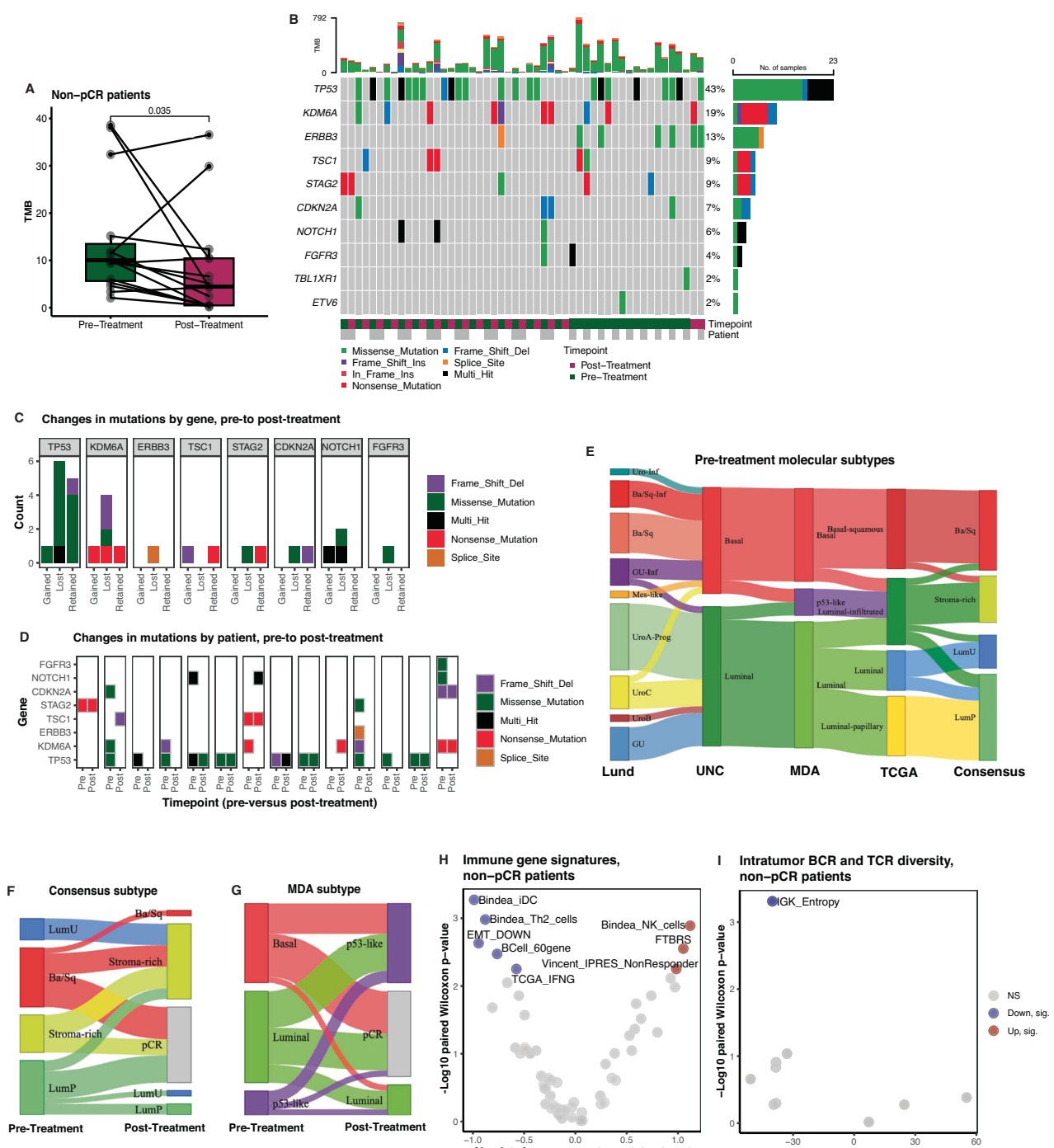

**Fig. 2 | MIBC tumors change with NAC-ICI. A** TMB in pre-treatment and post-treatment tumors from non-pCR patients (paired two-sided Wilcoxon test; $n = 15$). **B** Oncoplots of the 10 most common mutations in pre-and post-treatment tumors compared between timepoints ($n = 51$ samples). **C** Mutations in non-pCR tumors compared between pre-and post-treatment ($n = 28$ samples). **D** Mutations gained, lost, or retained from pre-to post-treatment in non-pCR tumors ($n = 28$ samples). **E** Pre-treatment molecular subtype by patient according to 5 subtyping schemes ($n = 37$). **F** Change in Consensus subtype among non-pCR patients from pre-treatment to post-treatment ($n = 37$). **G** Change in MDA subtype among non-pCR patients from pre-treatment to post-treatment ($n = 37$). **H** Changes in tumor RNA-based immune gene signatures from pre-to post-treatment among non-pCR patients (two-sided Wilcoxon test; $n = 37$). **I** Relative changes (differences divided by pre-treatment mean) in TCR and BCR diversity metrics among non-pCR patients (two-sided Wilcoxon test; $n = 37$). For all box-plots: centre = median; lower bound of box = 25th percentile; upper limit of box = 75th percentile; lower whisker = minimum value, 25th percentile − 1.5*IQR; upper whisker = maximum value, 75th percentile + 1.5*IQR.

cancer predictive biomarkers of response to ICI[27]. TMB decreased between patient-matched pre-treatment and post-treatment tumors (Fig. 2A), suggesting the possibility that immune editing–tumor evolution occurred[28]. The decrease in TMB was not correlated with changes in tumor purity estimated from pathology (Supplementary Fig. 2A) or tumor cellularity calculated by Sequenza (Supplementary Fig. 2B, C). Furthermore, the decrease in TMB does not appear to be confounded by reduced detection of mutations in small tumors, because the number of mutations in paired pre-and post-treatment NR and PR tumors is not significantly different at post-treatment

(Supplementary Fig. 2D), and the difference between median read depth in pre-treatment (149.5) and post-treatment (136) tumors at shared mutational loci is small (Supplementary Fig. 2E). 6 of 14 tumors lost mutations in the most frequently mutated gene, *TP53* (Fig. 2B–D). Next, we analyzed changes in tumor RNA-sequencing-based molecular subtypes (Fig. 2E). 10/18 tumors switched to a stroma-rich Consensus subtype (Fig. 2F), and 11/18 tumors switched to a p53-like MDA subtype (Fig. 2G), consistent with chemotherapy-induced subtype switching reported by Seiler et al[29]. The observed subtype switching does not appear to be confounded by changes in tumor cellularity (Sequenza) or tumor purity between pre-and post-treatment samples (Supplementary Fig. 1).

To assess whether treatment altered gene expression in the tumor immune microenvironment, we analyzed changes in previously defined immune gene signatures[24,30] (Fig. 2H). From pre-to post-treatment, signatures for Innate anti-PD-1 Resistance non-response (IPRES_NonResponder), fibroblast TGF-beta response (FTBRS), and natural killer cells (Bindea_NK_cells) increased, while signatures for immature dendritic cells (Bindea_iDC), T helper 2 cells (Bindea_Th2_cells), decreased epithelial-mesenchymal transition (EMT_DOWN), B cells (BCell_60gene), and IFNγ (TCGA_IFNG) decreased (FDR-corrected $p < 0.05$). We also assessed changes in inferred intratumoral BCR and TCR diversity metrics and observed decreased IGK entropy from pre-to post-treatment (FDR-corrected $p = 0.011$; Fig. 2I).

To measure systemic effects of treatment, we analyzed immune populations and plasma analyte concentrations in peripheral blood sampled at pre-treatment, pre-administration of cycle 2 (C2), and post-treatment. From pre-treatment to pre-cycle 2, T cell populations exhibited less PD-1 positivity, consistent with anti-PD-1 treatment (Fig. 3A). CD4+ Helios + , CD4 + TIGIT + CD39 + , and CD8 + CD39 + T cells increased, showing expansion of anti-inflammatory or regulatory populations[31]. CD39 expression may characterize neoantigen reactive T cells[32,33] and populations more likely to respond to ICI[34]. Additionally, transitional B cells (Lineage- CD19 + CD10 + ) and two myeloid-derived suppressor cell (MDSC) populations decreased. In the plasma analytes, only CXCL9, a chemotactic cytokine for T cells in the tumor micro-environment (TME)[35], increased between pre- and post-treatment samples (Fig. 3B), but many pro-and anti-inflammatory analytes increased from pre-treatment to pre-cycle 2 (Fig. 3C). IL-9 increased in responders compared with non-responders (no pathologic response) from pre-treatment to pre-cycle 2 (FDR-corrected $p = 0.047$; Fig. 3D). Overall levels of IL-9 decreased from pre-treatment to pre-cycle 2 in patients who did not respond (Fig. 3E). The change in IL-9 during the first cycle of NAC-ICI is a strong predictor of response ($AUC = 0.83$, $p = 0.001$; Fig. 3F). In a logistic regression model, changes in IL-9 concentration were significantly associated with response (logistic regression: $p = 0.015$; likelihood ratio test versus null: $p = 0.001$). Finally, we assessed changes in T and B cell receptor diversity from pre-to post-treatment and found no significant changes (Supplementary Fig. 2F).

## Clinical correlates of pathologic response

Response to immunotherapy depends on specific T cell targeting of tumor antigens[36]. We expected a subset of the tumor antigens to be driving immunotherapy response in the chemo-immunotherapy regimen, so we analyzed the landscape of predicted antigens in pre-and post-treatment tumors. Pre-treatment tumors exhibited a variety of predicted antigens from multiple sources, including predicted antigens from endogenous retroviruses (ERVs), indels, fusion events, single-nucleotide variants (SNVs), and self-antigens (cancer testis antigens; Fig. 4A). However, the number of predicted antigens per patient did not track closely with response (Supplementary Fig. 4B), TMB, Ayers IFNG signature, or CD8 T cell signature (Fig. 4A). We further assessed the number of shared predicted antigens because these could be attractive targets for development of biomarkers or vaccines

that could help multiple patients. While none of the predicted non-self-antigens were shared between patients, several predicted self-antigens were shared between multiple patients (Fig. 4B). Most antigens were lost or retained from pre-to post-treatment, with few antigens gained across treatment; however, there was no response association with type or count of antigens lost, retained, or gained across treatment (Fig. 4C). Several of the predicted antigens passed the more stringent antigen criteria of binding affinity, binding stability, and expression from Wells et al. (Fig. 4D–F). To assess whether certain mutational etiologies are targeted during chemo-immunotherapy response, we analyzed the COSMIC signatures of SNV and indel mutations and found a large proportion of the mutational landscape in both the pre- and post-treatment samples is attributable to APOBEC activity (Fig. 4G). We then predicted the most likely mutational signature for each SNV and indel mutation. While there were no clear associations between mutational signature and response, a current-smoker non-responder (Patient 1) gained new tobacco-related mutations, and one responder (Patient 2) gained platinum chemotherapy mutations during treatment (Fig. 4H).

Next, to identify which patients would benefit most from chemo-immunotherapy, we assessed the associations of pre-treatment clinical and immunogenomic features with patient outcomes. We first analyzed the response associations of pre-treatment TMB, RNA expression of 6 immune checkpoints (*PD-1, PD-L1, TIGIT, LAG3, TIM3*, and *VISTA*), and Ayers IFNG signature. Surprisingly, pre-treatment TMB was lower in partial responders than in non-responders (Fig. 5A). This effect could be due to the interaction between TMB and the occurrence of an adverse treatment-related event that required cessation of chemotherapy, since patients who went on to have an adverse treatment-related event that required cessation of chemotherapy had higher pre-treatment TMB (Fig. 5B, Supplementary Fig. 5A). Overall, pre-treatment TMB was not significantly correlated with Ayers_IFNG signature (Supplementary Fig. 5B). Of the immune checkpoints, only two were differentially expressed between response groups, with *PD-L1* (Fig. 5C) and *TIGIT* (Fig. 5D) higher in complete responders than in non-responders. *PD-L1* expression and Ayers_IFNG signature were significantly correlated with each other (Supplementary Fig. 5C), and CD8 infiltration (Bindea_CD8_T_cells) was significantly associated with many of the other immune gene signatures (Supplementary Fig. 5D). Ayers_IFNG was lower in partial responders than in non-responders (Fig. 5E). We did not observe significant differences in pre-treatment Consensus subtype by response group (Fig. 5F). While response was strongly associated with recurrence-free survival[9], over 25% of non-responders survived beyond 36 months, thus we asked which features were associated with survival in the non-responders. We found that male non-responders survived longer than female non-responders (Fig. 5G). The effect of gender on survival by response was not observed in a comparison data set of metastatic urothelial cancer treated with ICI (Supplementary Fig. 5E).

To summarize the independent effects of pre-treatment immune gene signatures, immune checkpoint RNA expression, TMB, and clinical variables in response and survival, we performed elastic net multivariable regression with 10-fold internal cross-validation. The variables with the largest coefficients associated with better response were Bindea CD8 T cell signature, stroma-rich Consensus subtype, and IL-8 signature, while the variable with the largest coefficients associated with worse response were Bindea iDC signature and current smoker (Fig. 5H). Using the mean coefficient values from 10-fold cross-validation, the response model predictions were strongly associated with response (Fig. 5I). The variables with the largest coefficients associated with better survival were male gender, stroma-rich Consensus subtype, and Bindea eosinophil signature, and the variables with the largest coefficients associated with worse survival were baseline ECOG performance status of 0, Macrophages signature, and Martinez Gordon M2 signature (Fig. 5J).

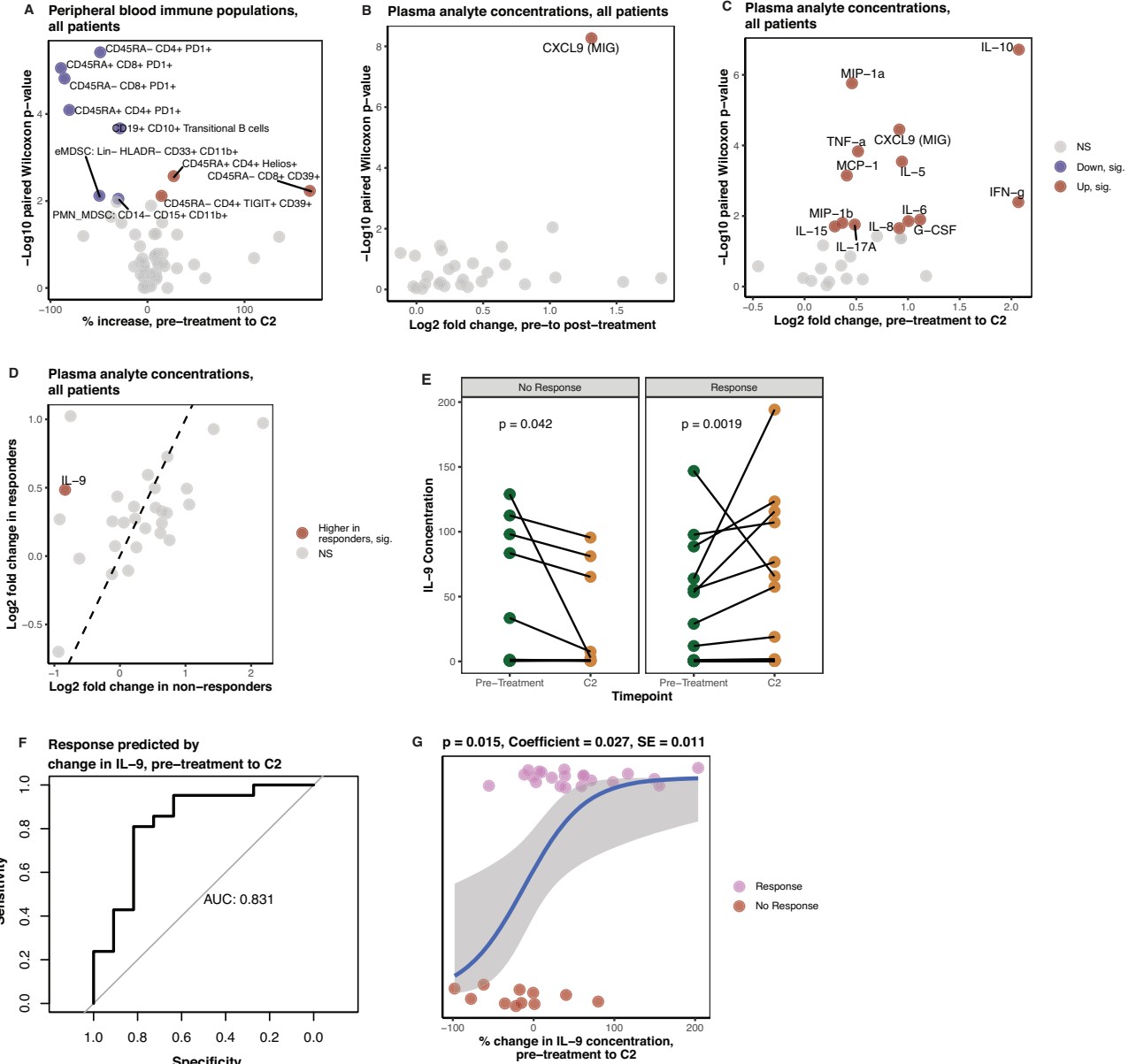

**Fig. 3 | Immune populations and plasma analyte features in peripheral blood change with NAC-ICI. A** Changes in peripheral blood flow cytometry markers from pre-treatment to pre-cycle 2 (n = 28; two-sided paired Wilcoxon test). **B** Changes in plasma analyte concentrations from pre-treatment to post-treatment (n = 37; two-sided paired Wilcoxon test). **C** Changes in plasma analyte concentrations between pre-treatment and pre-cycle 2 (n = 32; two-sided paired Wilcoxon test). **D** Changes in plasma analyte concentrations between pre-treatment and pre-cycle 2 in responders (n = 11) versus non-responders (n = 21). **E** Changes in IL-9 concentrations from pre-treatment to pre-cycle 2 in responders (n = 11) and non-responders (n = 21, two-sided paired Wilcoxon tests, no FDR correction). **F** Receiver operator characteristic curve of change in IL-9 concentration from pre-treatment to pre-cycle 2 versus response (n = 32). **G** Logistic regression of change in IL-9 concentration versus response. Error bands span the 95% confidence interval (n = 32).

## Comparison of NAC-ICI to neoadjuvant ICI

From a clinical perspective, it will be important to identify which patients who are eligible for both chemotherapy and immunotherapy would benefit from the combination. To investigate which patients benefited from combination chemo-immunotherapy versus immunotherapy alone, we compared data from LCCC1520 to ABACUS[7], a trial of neoadjuvant atezolizumab, an anti-PD-L1 agent. First, we assessed the associations of clinical, immune gene signatures, TMB, Consensus subtypes, and immune checkpoint RNA expression with response in LCCC1520 versus ABACUS. IL-8 signature and stroma-rich subtype were associated with better response to NAC-ICI compared to ICI alone (Fig. 6A). Patients with high pre-treatment IL-8 signature had

a higher response rate to NAC-ICI, while patients with low pre-treatment IL-8 signature had similar response rates with NAC-ICI or ICI alone (Fig. 6B–D). In patients treated with NAC-ICI, tumor IL-8 signature was associated with the eMDSC immune population in the peripheral blood (FDR-corrected p = 0.007; Supplementary Fig. 6A-B) and with MDSC signature[37] in the tumor (Supplementary Fig. 6C). Among patients with pre-treatment stroma-rich Consensus subtype, a higher response rate was observed in patients receiving chemo-immunotherapy (6/7 patients) versus immunotherapy alone (1/8 patients), while among patients with other pretreatment Consensus subtypes, similar survival was observed in patients receiving NAC-ICI or ICI alone (Fig. 6E–F).

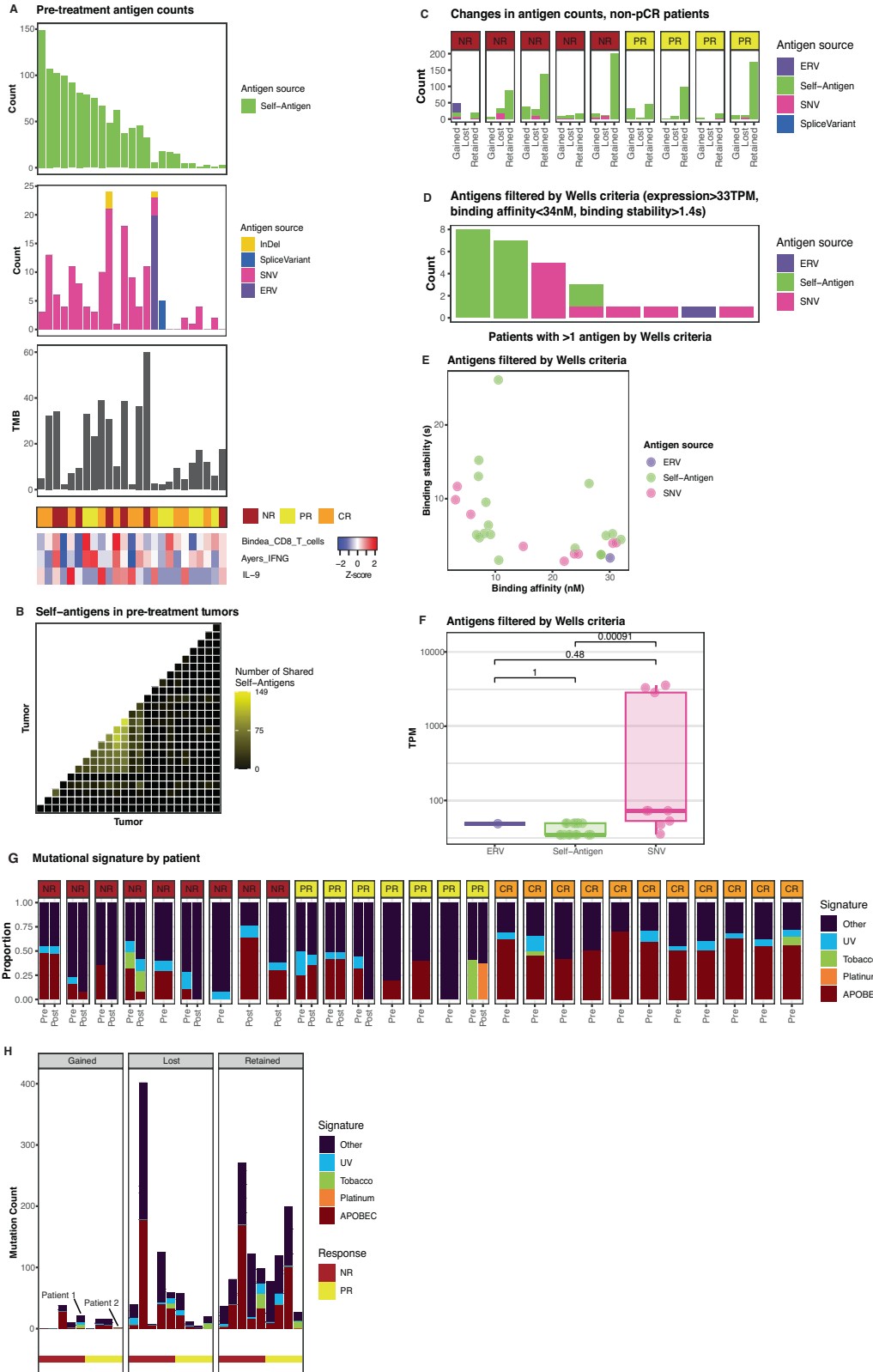

**Fig. 4 | MIBC tumors exhibit varied predicted antigen landscapes. A** Counts of predicted antigens (binding affinity < 500 nM) in pre-treatment tumors (*n* = 25). **B** Number of predicted self-antigens shared between pre-treatment tumors. **C** Counts of predicted antigens lost, retained, or gained from pre-to post-treatment in non-pCR tumors (*n* = 9). **D** Counts of predicted antigens in pre-treatment tumors filtered based on Wells criteria (*n* = 25). **E** Binding affinity and stability of Wells criteria-filtered predicted antigens (*n* = 25). **F** TPM expression levels of Wells criteria-filtered predicted antigens: ERV (*n* = 1 antigen), Self-antigen (*n* = 17

antigens), and SNV (*n* = 9 antigens). **G** COSMIC signatures of mutations in pre-and post-treatment tumors (*n* = 27 patients). (**H**) Counts of mutations lost, gained, and retained from pre-to post-treatment in non-pCR tumors, labeled by COSMIC signature (*n* = 9 patients). For all boxplots: centre = median; lower bound of box = 25th percentile; upper limit of box = 75th percentile; lower whisker = minimum value, 25th percentile − 1.5*IQR; upper whisker = maximum value, 75th percentile + 1.5*IQR. Unless otherwise noted, all pairwise comparisons are two-sided Wilcoxon tests.

## Discussion

We describe here potential biomarkers of response to chemo-immunotherapy in the phase II trial of neoadjuvant gemcitabine and cisplatin plus pembrolizumab in patients with MIBC. In this paper, we identify molecular and cellular features associated with outcome including a comparison of neoadjuvant chemo-immunotherapy with immunotherapy and chemotherapy alone.

Several key findings warrant future investigation. First, we observed tumor molecular subtype switching from pre-treatment to post-treatment, with many tumors switching to Consensus stroma-rich, MDA p53 subtypes, an effect not associated with changes in tumor cellularity. A similar pattern of subtype switching has been observed by Seiler et al in response to neoadjuvant chemotherapy, resulting in the CCR4-Scar-like phenotype[38]. Subtype switching may be caused by phenotypic transcriptomic changes, or by clonal expansion of one subtype versus another in a heterogeneous tumor[38]. Seiler et al found that patients with CCR4-Scar-like tumors went on to have the most favorable prognosis. Because many of the tumors in LCCC1520 switched to a Stroma-rich subtype, perhaps similar to the CCR4-Scar-like phenotype assessed by Seiler et al, we hypothesize that subtype switching on neoadjuvant chemo-immunotherapy could improve patient outcomes post-therapy.

Second, we observed a set of genetic variants in tumors pre-and post-treatment. The most frequently mutated genes in the LCCC1520 pre-treatment tumors were TP53 and KDM6A, which are two of the most commonly mutated genes in muscle-invasive bladder cancer[39]. In the 14 non-pCR patients with paired pre-and post-treatment samples, we observed an abundance of TP53 (10/14) and KDM6A (5/14) mutations in the pre-treatment tumors. However, 6 of 10 TP53-mutant tumors lost TP53 mutations between pre-and post-treatment, and 4/5 KDM6A-mutant tumors lost their KDM6A mutations between pre-and post-treatment, suggesting that immune editing could have occurred. Alternatively, treatment-induced clonal selection could have occurred: TURBT could be a bottleneck event, where there is only clonal outgrowth from a small population of a heterogeneous original tumor.

In addition to tumor features, we also observed changes in the immune markers in the peripheral blood across treatment. By flow cytometry, we observed decreased PD-1 + T cells, increased CD39 + T cells, and decreased MDSCs. The decrease in PD-1 + T cells could be due to pembrolizumab interfering with the PD-1 flow cytometry antibody, but anti-PD-1 has also been shown not to interfere with the specific flow cytometry antibody used[40]. CD39, along with its role as a suppression and exhaustion marker, also has been shown to delineate tumor-reactive CD8 + T cells and associates with response to ICI but not chemotherapy[34]. Thus, the increase in CD39 + T cells could correspond with an increase in tumor-reactive T cells in ICI treatment but we did not specifically see an increase in CD8 + , CD39 + T cells in responders. The decrease in MDSCs in the peripheral blood is consistent with a treatment-driven reduction in suppressive populations. Second, we found increases in the concentration of plasma cytokine IL-9 in responders versus non-responders from pre-treatment to cycle 2 of treatment. IL-9 can stimulate an anti-tumor response by Th9 cells[41] and has been shown to improve ICI response in a mouse model of lung cancer[42]. Plasma IL-9 is potentially promising for testing as a predictive non-invasive biomarker for response to NAC-ICI. IL-9 was the only plasma analyte whose change from pre-treatment to pre-cycle 2 was significantly associated with response, linking a change in a peripheral blood population with the therapeutic effect on the tumor. We also observed significant overall increases in multiple other analytes, including IL-8, IL-10, IFN-γ, and TNF-a, which could have stimulatory or inhibitory immune effects.

We also assessed sequencing-based features associated with outcomes. Using the LENS antigen prediction platform, we were able to predict thousands of antigens from different sources and compare these antigens head-to-head by immunogenicity features and by treatment timepoint. We analyzed shared antigens because these are attractive targets for development of biomarkers or vaccines that could help multiple patients. We did not find any associations between antigen types or counts and response. While we cannot yet predict well which antigens are most likely to induce an anti-tumor T cell response, these data could help in future efforts to understand tumor antigen immunogenicity. We did, however, see strong associations of two pretreatment tumor features–IL-8 signature and Consensus stroma-rich subtype–with improved response to chemo-immunotherapy versus a previously published study where patients received atezolizumab without chemotherapy (ABACUS), suggesting that these features are associated with the added response benefit of incorporating neoadjuvant chemotherapy with immunotherapy. IL-8 is an immunosuppressive cytokine rich in the tumor microenvironment that is associated with worse ICI response[43,44] and recruitment of immunosuppressive MDSCs to the tumor[45]. Chemotherapy can decrease specific immunosuppressive myeloid populations, including MDSCs[46], suggesting that the response benefit in chemo-immunotherapy in IL-8 signature-high tumors could be due to the effects of chemotherapy on cells that inhibit ICI response. If IL-8 is driving tumor immune resistance, then adding chemotherapy to immunotherapy could provide differential benefit. Consistent with this notion, tumor IL-8 signature is correlated with the percentage of the peripheral blood eMDSC flow cytometry population, and from pre-treatment to pre-cycle 2, both the eMDSC and PMN-MDSC population percentages decreased. Tumor stroma and stromal tumor-infiltrating lymphocytes also help mediate bladder cancer response[47], so the increased response in the stroma-rich bladder cancer subtype could be associated with effects of chemotherapy on the stromal compartment that improve ICI response. We have previously shown in a stroma-rich preclinical model that nanoparticle delivered gemcitabine and cisplatin are able to decrease alpha-SMA positive cancer associated fibroblasts[48]. While it is impossible to discern definitively the benefit of the immunotherapy in NAC-ICI, this manuscript provides evidence that immunotherapy could have effects (e.g. decreased % PD-1 + CD8 T cells), and these effects are consistent with improved tumor response.

Several other potential biomarkers associated with outcome. Females had worse survival among non-responders. While there are too few females in the trial to assess factors influencing sex-specific differential survival by response, these data suggest that a potential biomarker of response might differ by sex. Additionally, iDC and macrophages were associated with worse outcomes in our multivariable model. iDCs[49] and tumor-associated macrophages[50] both have potential roles in attenuating tumor responses to ICB.

We also made several findings of unknown significance. First, we observed increases in FTBRS signature and decreases in TCGA_IFNG signature across treatment. High FTBRS signature[4] and low TCGA_IFNG[51] have been associated with worse response to ICI, suggesting that the post-treatment tumors might respond worse to any future ICI treatment than if they had not been treated with NAC-ICI. Second, we observed decreased IGK entropy from pre-to post-treatment but no significant changes in BCR or TCR diversity. While CDR3 sequences from IGK are abundant (54% of the B cell population), no studies to our knowledge have documented that decreases in the entropy of IGK are associated with NAC-ICI therapy. Third, we observed that pre-treatment TMB was not significantly correlated with Ayers_IFNG signature, consistent with prior literature demonstrating poor correlation between mutational burden and cytolytic signatures[52,53]. While both TMB and Ayers_IFNG signature have been proposed as potential predictive biomarkers of ICI response[18,54], we report differential associations of the two markers with NAC-ICI response. Further clinical correlative analysis from future clinical trials will need to be conducted to contextualize and to evaluate the generalizability of these findings.

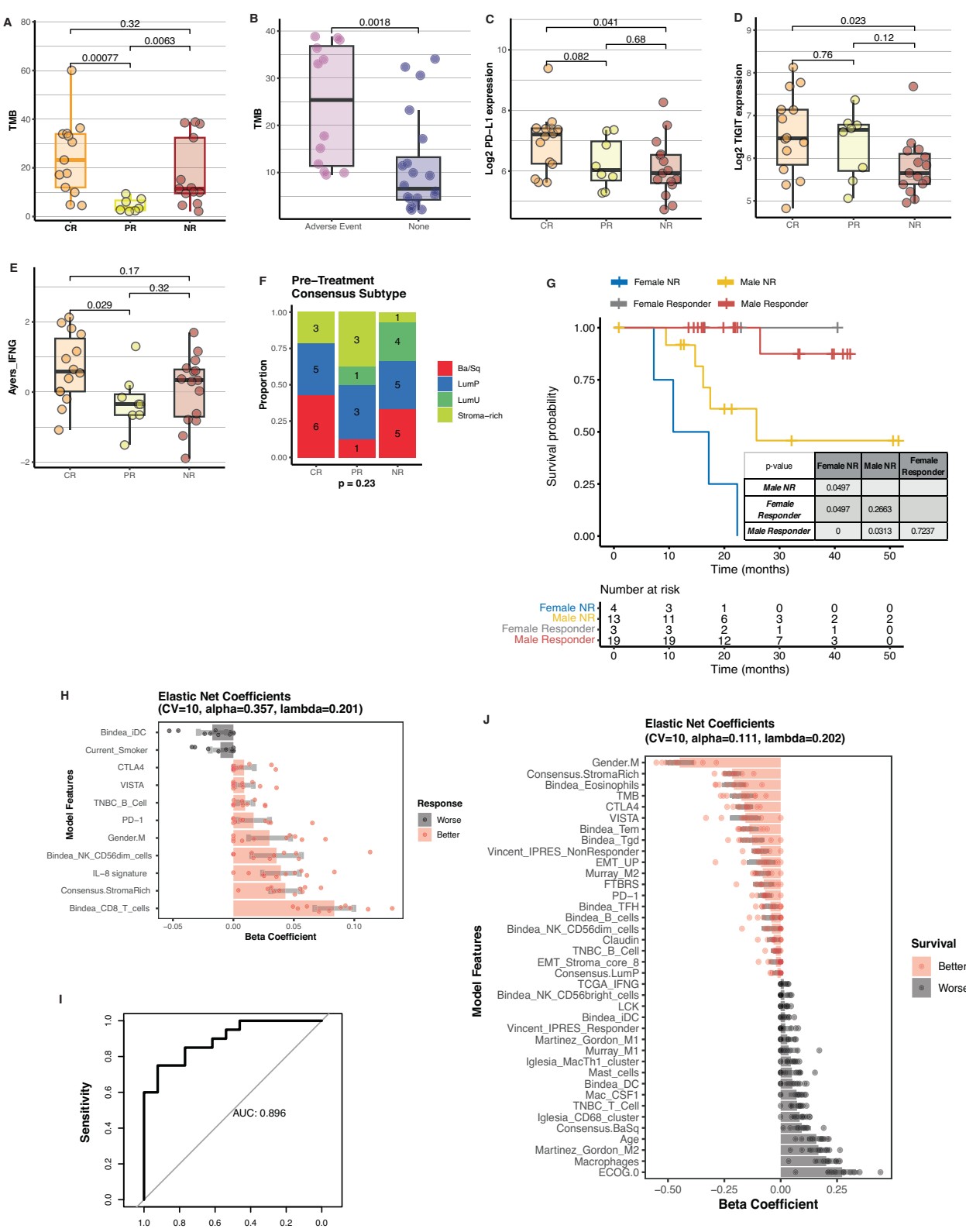

While analysis of the correlative data from this trial of neoadjuvant chemo-immunotherapy presents interesting findings, it has several limitations. The trial is small and has limited data for certain analyses, increasing the likelihood of spurious findings. Validation on separate chemo-immunotherapy cohorts in the future will be necessary. While analysis of pathologic response is confounded by the fact that TURBT removes much or all of the tumor, our data adds evidence that there are immune-related predictors of response to TURBT plus NACI-ICI, suggesting that not only TURBT but also the systemic treatment associated with longer survival. The comparison immunotherapy-only trial, ABACUS, used atezolizumab ICI instead of pembrolizumab, and while both are monoclonal antibody therapies that target the PD-1 axis, the two drugs might have important functional differences. The LCCC1520 and ABACUS patient populations are also different, with a

**Fig. 5 | Clinical and immunogenomic features are associated with MIBC patient outcomes. A** TMB levels from pre-treatment tumors are compared by patient response status (*n* = 33 patients: 13 CR, 7 PR, 13 NR; two-sided Wilcoxon tests). **B** TMB levels by an adverse treatment-related event that required cessation of chemotherapy (*n* = 32 patients: 12 CR, 7 PR, 13 NR; two-sided Wilcoxon test). **C–E** RNA expression of *PD-L1* and *TIGIT*, and Ayers IFNG immune gene signature values, from pre-treatment tumors are compared by patient response status (*n* = 37 patients: 14 CR, 8 PR, 15 NR; two-sided Wilcoxon tests). Groups are compared by Wilcoxon *p*-value. **F** Consensus subtype of pre-treatment tumors by response class (Fisher's exact test, no FDR correction; *n* = 37 patients). **G** Kaplan-Meier survival curves of female and male patients by response status. The difference in survival between female and male responders is compared by log-rank *p*-value. **H** Cross-validated elastic net coefficients of response, with only the feature coefficients shown that have 95% confidence intervals from cross validation that do not span zero (*n* = 33 patients). Points represent feature beta coefficients in each of 10 folds of cross validation. Bars represent mean beta coefficient value for each feature across 10-fold cross validation. Gray error bars represent 95% confidence intervals for each feature. **I** Receiver operating characteristic curve of model predictions on the training set. **J** Cross-validated elastic net coefficients of survival, with only the feature coefficients shown that have 95% confidence intervals from cross validation that do not span zero (*n* = 33 patients). Points represent feature beta coefficients in each of 10 folds of cross validation. Bars represent mean beta coefficient value for each feature across 10-fold cross validation. Black error bars represent 95% confidence intervals for each feature. For all boxplots: centre = median; lower bound of box = 25th percentile; upper limit of box = 75th percentile; lower whisker = minimum value, 25th percentile − 1.5*IQR; upper whisker = maximum value, 75th percentile + 1.5*IQR. Unless otherwise noted, all pairwise comparisons are two-sided Wilcoxon tests.

younger age (median: 66 vs. 73 years) and higher percentage of patients who have smoked (85% vs 78%) in LCCC1520 versus ABACUS. It is difficult to compare trial response rates head-to-head because of cohort and drug differences, but we compared differential response within the trials. A prospective randomized trial evaluating the different neoadjuvant treatment approaches with integrated biomarker analyses will help to better define predictive biomarkers for response and other clinical outcomes.

In summary, plasma IL-9, tumor IL-8 gene signature levels, and tumor stroma-rich subtype represent potential biomarkers of response to NAC-ICI. The eventual goal is to predict which patients would have increased or decreased survival after chemoimmunotherapy to aid patient selection. Future efforts must include further independent biomarker discovery and validation, included within the context of prospective clinical trials, ultimately to improve the selection of patients for ICI-related treatments.

## Methods

### Consent and IRB approval
Our research was reviewed and approved by the institutional review boards at Duke University (Duke University Health System Institutional Review Board) and UNC (UNC Institutional Review Board) and conducted in accordance with the ethical criteria set by the Declaration of Helsinki. All participants provided written informed consent for the use of their tissue or peripheral blood for these correlative assessments.

### Clinical trial design
Study Protocol can be found in Supplementary Information. Patients were enrolled between June 2016 and March 2020. Power calculations and statistical tests comply with ICMJE guidelines on reporting and can be found in Supplementary Information under "Sample Size, Power Calculations, and Accrual". Link to the NCT02690558 preregistered protocol (September 25, 2018): https://cdn.clinicaltrials.gov/large-docs/58/NCT02690558/Prot_SAP_000.pdf.

### Patient cohort
We analyzed correlative samples from the LCCC1520 phase II trial of neoadjuvant pembrolizumab and gemcitabine and split dose cisplatin chemotherapy in patients with MIBC prior to RC. In this study, 39 patients were treated with pembrolizumab 200 mg IV on day 1 in combination with gemcitabine 1 g/m² on day 1 and day 8, and cisplatin 35 mg/m² on day 1 and day 8 every 21 days for 4 total cycles. The first 6 accrued patients received one extra dose pembrolizumab 2 weeks prior to C1D1, and subsequent cisplatin at full dose (70 mg/m²) on day 1 prior to a protocol amendment (Fig. 1A). Tumor tissue was collected pre-treatment from transurethral resection of bladder tumor (TURBT) samples and post-treatment radical cystectomy samples. Blood was collected for peripheral blood mononuclear cells (PMBC) and plasma pre-treatment, cycle 2 day 1, and prior to RC.

### Response criteria
We defined response as <pT2N0 at cystectomy. We divided response into partial response (>pT0 at cystectomy) and complete response (pT0 at cystectomy).

### Tumor purity analysis
Tumor purity was assessed for 5 tumors with matched pre-and post-treatment samples. An expert pathologist [SW] estimated tumor counts (papillary and invasive) as a percentage of total cells on formalin-fixed paraffin-embedded slides stained with Hematoxylin & Eosin.

### RNA/DNA isolation from bladder tumors
RNA and DNA were extracted from formalin-fixed, paraffin-embedded (FFPE) tumor tissues at the UNC Lineberger Comprehensive Cancer Center Translational Genomics Lab (TGL). Genomic DNA and RNA were extracted on the Promega Maxwell 16 MDx instrument using the Promega Maxwell 16 FFPE Plus LEV DNA or RNA purification kits, respectively. DNA and RNA quantity and quality were assessed with the Qubit Flex Fluorometer (ThermoFisher) and the 4200 TapeStation system (Agilent), respectively. The nucleic acids were stored at −80 °C until used in DNA and RNA sequencing library preparations.

### Whole-exome sequencing of tumors
Pre- and post-treatment sequencing libraries were prepared from 100 ng of DNA using the Illumina DNA Prep with Enrichment kit and the Illumina Exome Panel. During library preparation, DNA was amplified with IDT for Illumina DNA Unique Dual Indexes, resulting in a pool of twelve dual-indexed paired-end libraries that were sequenced on Illumina's NovaSeq 6000 (2 × 100) version 1.5.

### TMB calculation
Realigned BAM files from Abra were passed through Strelka, Mutect2, and Cadabra. High confidence variants were kept. To calculate the denominator for TMB, samtools provided estimates of coverage depth at each exome location and bases adequately covered by reads were counted.

### Neoantigen prediction
Predicted neoantigens were identified using the Landscape of Effective Neoantigens Software (LENS) with default parameters[55]. Antigens were detected from single nucleotide variants (SNVs), insertions and deletions (InDels), splice variants, gene fusions, viruses, and human endogenous retroviruses (hERVs). Both SNVs and InDels were detected and filtered using the same workflow and parameter set. Specifically, tumor and normal WES reads were trimmed using TrimGalore! v0.6.2 and mapped to the hg38 human genome reference using bwa mem v0.7.17. RNA reads were also trimmed using Trim Galore! v0.6.2 and aligned to the hg38 human reference genome using STAR v2.7.0 f. DNA

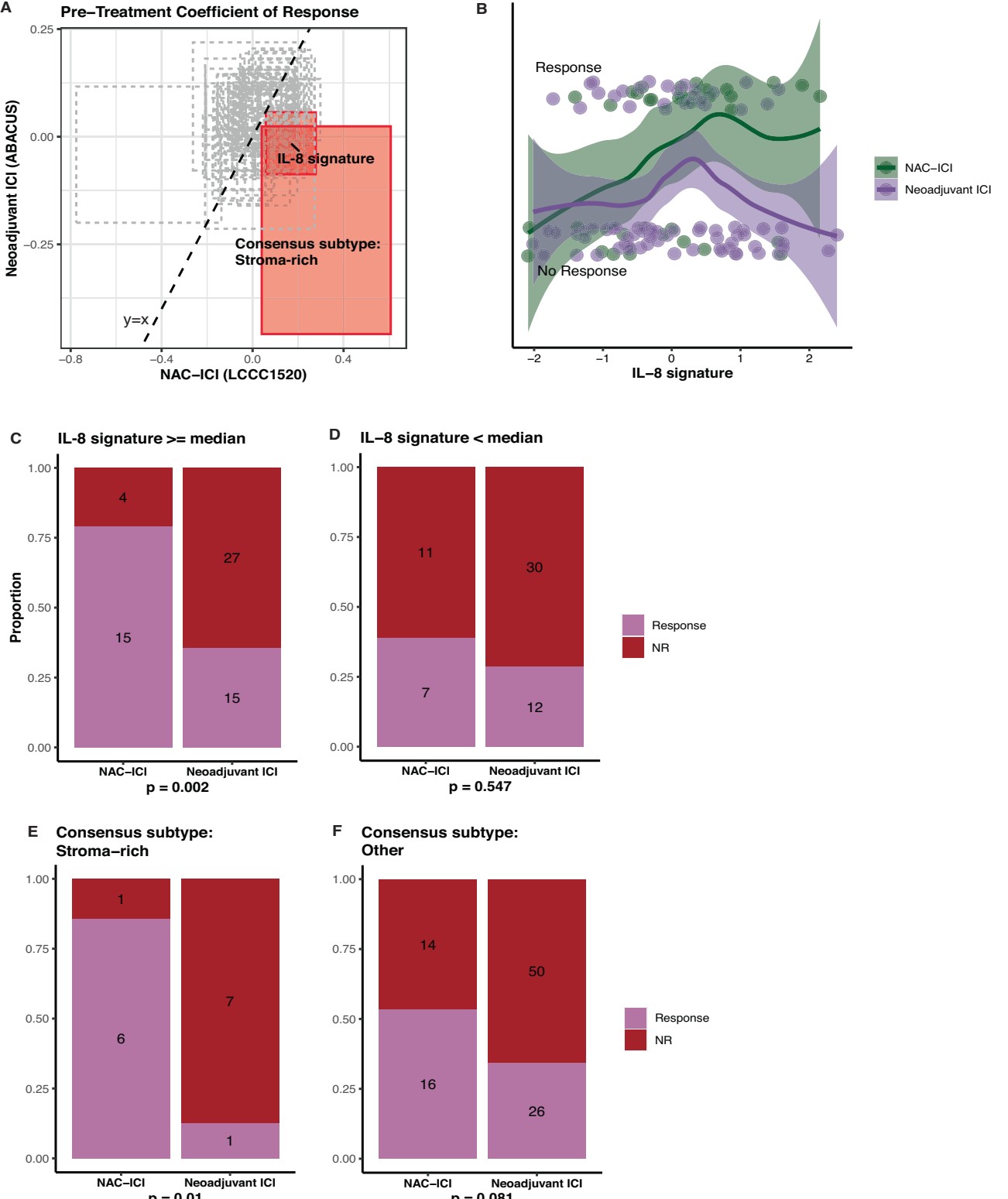

**Fig. 6 | NAC-ICI could improve outcomes compared to neoadjuvant ICI in specific subsets of patients. A** Comparison of association of pre-treatment immune gene signatures with response in ABACUS (ICI only, $n = 84$) versus LCCC1520 (Chemo-ICI, $n = 37$). Each pre-treatment immune gene signature's association with response is plotted as a rectangle spanning the 83.4% confidence interval of the coefficient. IL-8 signature and stroma-rich Consensus subtype are highlighted as the only signatures with significantly different associations with response in the two data sets, with confidence intervals that do not cross the $y = x$ line. **B** Response status by IL-8 signature in the two data sets. Loess curves are plotted with error bands spanning the 95% confidence interval. **C, D** Response status by data set in patients with high or low pre-treatment IL-8 signature values (Fisher's exact test, no FDR correction). **E, F** Response status by data set in patients with Stroma-rich or other Consensus subtype (Fisher's exact test, no FDR correction).

alignments were sanitized with Picard v2.21.4 for duplicate marking, ABRA v2.20 for InDel realignment, and GATK v4.1.6.0 for base quality recalibration. Somatic variants were called using MuTect2 from GATK v4.1.6.0, Strelka v2.9.9, and ABRA v2.20. Variants from each tool were filtered for PASS status and intersected among the variant callers. The resulting variant set were annotated using snpEff with a custom annotation. Transcript quantifications were performed using Salmon v1.1.0 with a Gencode v37 annotation GTF. SNVs classified as missense by snpEff's annotations, were associated a transcript within the upper quartile of expression (TPM), had at least one tumor RNA read containing the variant, and had NetMHCpan v4.1b binding affinity < 500 nM were included in the LENS report. InDels classified as disruptive insertion, disruptive deletion, conservative insertion, conservative deletion, or frameshift, were associated with a transcript in the upper quartile of expression, had at least one tumor RNA read containing the variant, and had NetMHCpan v4.1b binding affinity < 500 nM were included in the LENS report. Splice variants were detected using NeoSplice v0.0.3, a splice tumor antigen discovery tool. Specifically, tumor-associated k mers were required to exceed a tumor expression value of 20 and required not to exceed of 4 in a tissue-matched normal RNA sample (kmer_search_bwt.py's --tumor_threshold and --normal_threshold parameters, respectively). Splice variants were also only considered if they originated from a transcript with a minimum coverage of 100 reads (kmer_graph_inference.py's --transcript_min_coverage parameter). Splice variant pMHCs from NeoSplice were included in the LENS report if the peptide's coding sequence was independently detectable within the tumor RNA sequencing reads and had a NetMHCpan v4.1b binding affinity <500 nM. Gene Fusion tumor antigens were detected using STARFusion v1.10.1 using the associated CTAT Trinity reference. Fusion-derived pMHCs were included in the LENS report if at least one read supported the peptide's coding sequence in the tumor RNA data and the pMHC had a NetMHCpan v4.1b binding affinity <500 nM. Viral tumor antigens were detected using a modified VirDetect workflow (Selitsky et al., 2020). Specifically, viruses with at least 25% coverage from the tumor RNA reads were considered for peptide generation. The list of potential peptides was generated using a sliding window of kmer sizes 8–11 across the viral coding sequence. Peptides that had at least one tumor RNA read supporting the peptide's CDS and had a NetMHCpan v4.1b binding affinity <500 nM were included in the LENS output. hERV tumor antigens were detected by augmenting the Gencode v37 annotation GTF with hERV coding sequences to obtain quantification through Salmon v1.1.0. hERVs are known to be expressed in normal tissues and these hERVs are unlikely to be suitable tumor antigens. We filtered hERVs to a subset with at least 2-fold higher expression in the tumor RNA sample over a normal, tissue-matched RNA sample. This was performed using EdgeR to account for library size bias through the counts per million (CPM) metric. Peptides that survived filtering were used to generate a list of peptides using the same sliding window approach as the viral workflow. Peptides that had their coding sequences detectable within the tumor RNA reads and had a NetMHCpan v4.1b binding affinity < 500 nM were included in the LENS report. Cancer testis antigens and self-antigens were derived using a list of candidate cancer testis genes from CTDatabase (http://www.cta.lncc.br/). Transcripts generated by these genes were quantified using STAR v2.7.0 f and Salmon v1.1.0 and those above the 95th percentile were included for downstream processing. A sliding window approach, identical to the viral and hERV workflows, was used to generate a peptide list. Peptides that had coding sequences detectable within the tumor RNA reads and with a NetMHCpan v4.1b binding affinity < 500 nM were included in the LENS report.

### RNAseq from tumor RNA
The SMARTer Stranded Total RNA-Seq kit v2 (Takara) was used to generate sequencing libraries from the tumor RNA (50 ng) as this kit generates strand-specific Illumina-compatible libraries from partially degraded nucleic acids. Paired-end sequencing was performed using the NovaSeq 6000 version 1.0 or 1.5 (2 × 50). All sequencing for this study was performed at the High Throughput Sequencing Facility (HTSF) at the University of North Carolina at Chapel Hill.

### Gene expression
Reads were aligned to the hg38 genome using STAR, and genes were quantified using Salmon. On log2 transformed upper-quartile normalized expression data, molecular subtyping was performed using the BLCAsubtyping and consensusMIBC R packages. Immune gene signatures were calculated as the z-score value of all genes in the signature. T cell receptor (TCR) and B cell receptor (BCR) repertoire diversity metrics were calculated using FastQC and Mixcr[56,57] from PBMCs as well as inferred from tumor sequencing data.

### RNA Isolation and RNAseq for CD3+ cells
Total RNA was isolated from bead selected CD3+ cells from patient PBMC using Qiagen's AllPrep DNA/RNA Mini kit per manufacturer's instructions. RNA quantity and quality were assessed with the Qubit 4 Fluorometer (ThermoFisher) and the 2200 TapeStation system (Agilent), respectively. RNAseq libraries were prepared with Illumina's Stranded mRNA Prep Ligation kit, starting with 100 ng of total RNA. This method uses oligo(dT) magnetic beads to capture mRNAs with polyA tails. Paired-end sequencing was performed using the NovaSeq 6000 version 1.0 (2 × 100).

### Immunophenotyping of peripheral blood mononuclear cells (PBMC)
PBMC immunophenotyping was designed, performed, and analyzed by Immune Monitoring and Genomics Facility, UNC Lineberger Comprehensive Cancer Center. Cryopreserved PBMCs were thawed in Dextran-Albumin (CSL Behring #44206-251-10) solution and washed with AIM-V CTS (Gibco #0879122DK)/5% of Human AB Serum (Gemini#100-512), following cell resuspension in 1X HBSS (Gibco #14175-095). Viable PBMC numbers were determined and distributed at 2 million cells per patient per timepoint per assay tube. Flow cytometry panels were used to analyze T cell (Supplementary Table 1), B cell (Supplementary Table 2), and myeloid (Supplementary Table 3) subsets. Samples were aliquoted and analyses performed separately for T, B, and myeloid cells. Intracellular staining for Foxp3 in T cell panel and Ki67 in B cell panel were performed by utilizing eBioscienceFoxp3/TF staining kit (ThermoFisher #00-5523-00). All stained cells were acquired on BD LSRFortessa (Serial# H64717700116). Gating strategy was based on fluorochrome minus one (FMO) control staining to distinguish positive and negative populations in each subset. All gating and subset analyses were done by FlowJo v8 (Supplementary Fig. 3). Flow cytometry values were standardized between aliquots by assessing parent population percentage. Percentages of parental populations were analyzed by paired Wilcoxon tests.

### Plasma analyte concentrations
Plasma protein levels were determined by multiplex immunoassay provided by Meso Scale Discovery (Rockville, MD). The Custom 28-Plex panel consisted of the 26 analytes in the U-PLEX Human Immuno-Oncology Group 1 as well as two additional R-PLEX assays. For each analyte measured, biotinylated capture antibodies were coupled to a U-PLEX Linker assigned a specific spot arrayed at the bottom of plate wells. Plasma samples were run in duplicate at 2-fold dilution. Detection antibodies utilized SULFO-TAG™ labeling, and the electrochemiluminescent signal was measured with the MESO QuickPlex SQ 120. Preliminary analyses and data organization were performed using MSD Discovery Workbench 4.0.

## Code

Data analysis and figure generation were performed using R v4.3.0. Packages used: ggplot2 v3.4.2, plyr v1.8.8, dplyr v1.1.2, maftools v2.16.0, flipPlots v1.3.6, ggpubr v0.6.0, pROC 1.18.2, ggcorrplot v0.1.4, gplots v3.1.3, scales v1.2.1, reshape2 v1.4.4, ggh4x v0.2.4, ggnewscale v0.4.9, glmnet v4.1.7, caret v6.0.94, caTools v1.18.2, survival v3.5.5, survminer v0.4.9, ggpmisc v0.5.5, matrixStats 1.0.0, ggrepel v0.9.3

## Elastic net modeling

TMB, gene expression of 6 immune checkpoints (*PD-1, PD-L1, TIGIT, LAG3, TIM3, CTLA4*, and *VISTA*), 61 immune gene signatures[24] (Supplementary Table 4), Consensus molecular subtype, and 5 clinical variables (age, T stage, ECOG performance status, gender, and smoking status) were evaluated for their associations with response and survival. Variables were standardized (mean = 0, SD = 1) and samples with any NA values were omitted. Elastic net regression with 10-fold cross-validation was used to build an optimal model of response using the R package caret (tuneLength = 10) and an optimal model of survival using the R package glmnet[58]. The β-coefficient mean and 95% confidence interval from the 10 folds were calculated for each predictor.

## Statistical analysis

Unless otherwise noted, groups were compared by Wilcoxon tests with Benjamini-Hochberg FDR correction, and proportions were compared by Fisher's exact test.

## ABACUS trial

A single-arm phase 2 study of neoadjuvant immune checkpoint therapy in muscle-invasive bladder cancer[7] (NCT02662309). In brief, patients with histologically confirmed T2-T4a transitional cell carcinoma of the bladder were treated with two 3 weekly cycles of atezolizumab pre-cystectomy. Tumor samples were harvested before (TURBT) and after (cystectomy) treatment. RNA was extracted from formalin-fixed paraffin-embedded tumor samples and sequenced using Illumina sequencing-by-synthesis.

## Reporting summary

Further information on research design is available in the Nature Portfolio Reporting Summary linked to this article.

## Data availability

LCCC1520 trial: The processed DNA and RNA sequencing data generated in this study have been deposited in the dbGaP database under accession code phs003452.v1.p1. The sequencing data are available under restricted access due to data privacy laws. Request access can be obtained through dbGAP. Access is permitted for 1 year. Individual de-identified patient clinical variables, besides patient age, are shared (Supplementary Data 1). Additional source data to reproduce main and Supplementary Figures. are available in Figshare [https://doi.org/10.6084/m9.figshare.23705790]. Additional individual de-identified participant data, including age, can be shared upon request to the corresponding authors. ABACUS trial: The raw sequencing data are available under restricted access in the European Genome-Phenome Archive[7] under accession EGAD00001006205. Data access can be granted via the EGA. The authors declare that all other data supporting the findings of this study are publicly available within the paper and its supplementary information files.

## Code availability

Code to generate the data and figures for this study is available as Supplementary Code 1.

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

## Acknowledgements

This work was supported by a grant to JSS from Merck Sharp & Dohme LLC, a subsidiary of Merck & Co., Inc., Rahway, NJ, U.S.A. (MSD). TLR is supported by the National Cancer Institute at the National Institutes of Health (grant number 1K08CA248967-01, TLR). WB is supported by the National Cancer Institute at the National Institutes of Health (grant number 1F30CA278317-01A1, WB).

## Author contributions

Study conception, planning, and sample acquisition: M.G.W., H.W., G.A., A.D.W., K.F., L.F., M.R.H., K.P.M., T.L.R., M.I.M., J.S.S., W.Y.K., B.G.V. Data analysis, interpretation, and manuscript writing and review: W.B., M.Z., J.S.L., S.P.V., M.G.W., H.W., S.E.W., G.A., A.D.W., J.S.D., K.P.M., T.L.R., M.I.M., J.S.S., W.Y.K., B.G.V.

## Competing interests

The authors declare the following competing interests: Pfizer (Stock and Other Ownership Interests); Loxo/Lilly (Consulting or Advisory Role); Merck, Roche/Genentech, Bristol-Myers Squibb, Mirati Therapeutics, Incyte, Seagen, G1 Therapeutics, Alliance Foundation Trials, Alliance for Clinical Trials in Oncology, Clovis Oncology, Arvinas, ALX Oncology, Loxo, Hoosier Cancer Network (Research Funding); Elsevier, Medscape, Research to Practice (Other Relationship).

## Additional information

**Wolfgang Beckabir[1,2], Mi Zhou [1,3], Jin Seok Lee[1,3,4], Steven P. Vensko[1], Mark G. Woodcock [1,5], Hsing-Hui Wang[1,2], Sara E. Wobker[1,6], Gatphan Atassi [1], Alec D. Wilkinson[1], Kenneth Fowler[1], Leah M. Flick[1], Jeffrey S. Damrauer [1,5], Michael R. Harrison[7], Karen P. McKinnon[1,2], Tracy L. Rose[1,5], Matthew I. Milowsky [1,5], Jonathan S. Serody [1,2,5,8,11] ✉, William Y. Kim [1,3,5,9,11] ✉ & Benjamin G. Vincent [1,3,4,5,8,10,11] ✉**

[1]Lineberger Comprehensive Cancer Center, University of North Carolina at Chapel Hill, Chapel Hill, NC, USA. [2]Department of Microbiology and Immunology, UNC School of Medicine, Chapel Hill, NC, USA. [3]Department of Genetics, University of North Carolina at Chapel Hill, Chapel Hill, NC, USA. [4]Curriculum in Bioinformatics and Computational Biology, UNC School of Medicine, Chapel Hill, NC, USA. [5]Division of Oncology, Department of Medicine, University of North Carolina at Chapel Hill, Chapel Hill, NC, USA. [6]Department of Pathology and Laboratory Medicine, University of North Carolina at Chapel Hill, Chapel Hill, NC, USA. [7]Division of Medical Oncology, Department of Medicine, Duke Cancer Institute, Duke University, Durham, NC, USA. [8]Department of Pharmacology, University of North Carolina at Chapel Hill, Chapel Hill, NC, USA. [9]Division of Hematology, Department of Medicine, UNC School of Medicine, Chapel Hill, NC, USA. [10]Computational Medicine Program, UNC School of Medicine, Chapel Hill, NC, USA. [11]These authors contributed equally: Jonathan S. Serody, William Y. Kim, Benjamin G. Vincent. ✉e-mail: jonathan_serody@med.unc.edu; wykim@med.unc.edu; benjamin_vincent@med.unc.edu

