## [Peer Review File · Nature Communications]

Immune features are associated with response to neoadjuvant chemo-immunotherapy for muscle-invasive bladder cancerREVIEWER COMMENTS

Reviewer #1 (Remarks to the Author): with expertise in bladder cancer (therapy, omics)

Manuscript ID: NCOMMS-23-31531-T entitled " Immune features are associated with response to neoadjuvant chemo-immunotherapy for muscle-invasive bladder cancer"

The authors investigated samples from patients that were treated within a clinical trial. Bladder cancer patients received neoadjuvant gemcitabine and cisplatin plus pembrolizumab. Tumor specimens were sampled at TURBT and when still present at cystectomy. In addition, blood was sampled before during and after neoadjuvant treatment. In tumor specimens, RNA seq and whole-exome sequencing was performed. In the blood specimens, immunphenotyping of PBMCs using flow cytometry. In addition, RNAseq was performed in T-cells from PBMCs. Finally multiplex immunoassays were used to determine levels of chemokines.

Several changes could be observed between timepoints and several parameters were associated with response. An increase of IL-9 plasma levels before and during treatment was associated with favorable response. Total mutational burden decreases before to after neoadjuvant treatment. 6 tumors lost TP53 mutations and the molecular subtype most frequently changed to a stroma-rich consensus subtypes. High expression levels of PD-L1 and TIGIT were associated with complete response. Finally, they identified IL-8 signature and stroma-rich subtype were associated with response to neoadjuvant chemo-immune checkpoint inhibition but not to checkpoint inhibition alone.

Comments and questions to the authors:

This is a relevant and comprehensive molecular investigation patients treated with neoadjuvant chemo- and immunotherapy. The data is clearly presented and the analysis are in line with clinically relevant questions and the conclusions are supported by the findings.

1. Do the authors have information on purity of tumor samples? If not, the reviewer suggests to add this information and compare to present data (e.g. molecular subtype, TMB, immune gene signatures).

2. The authors state that subtype switching was not associated with tumor cellularity but use gene expression data to underline this statement. The reviewer suggests to support this

statement by data on purity of samples or IHC.

3. The authors performed flow cytometry and included approx. 43 antibodies. Were samples aliquoted and analyses separate for T-, B- and myeloid cells? If so, then this should more clearly be described in the material and method section. In addition, how did the authors standardize flow cytometry values for analyses between different aliquotes (e.g. for Figure 3A)?

4. The authors further analysed features that were associated with survival in non-responders and identified that male patients survived longer. Which other parameters were investigated? This information could e.g. be give in an extended table.

5. Do the authors see other options, why tumors lost TP53 and KDM6A under treatment? They suggest immune editing, could treatment induced clonal selection play a role?

6. The authors performed elastic net modeling to evaluate features for their associations with response. Did the authors try to build a model based on all of these parameters to predict response (e.g. random forest, GLM or similar)? Internal crossvalidation (e.g. leave one out) would allow to get information on the performance of such a model. The accuracy of such a model, trained on a comprehensive panel of biomarkers might be of interest for researcher and help to guide biobanking for further clinical trials.

7. The conclusions of the abstract and the summary of the main manuscripts are quite different. Both seem to be appropriate but the reader gets different information when reading only one or the other. The reviewer suggests to modify either one or the other.

Minor:

- Results, clinical features of NAC-ICI, 2nd line: (Rose et al), other refs are given in numbers in superscript.

- The CCR4-Scar-like subtype is not described in the here given reference.

Reviewer #2 (Remarks to the Author): with expertise in bladder cancer, (immuno)therapy

In their manuscript, Beckabir et al studied a cohort of MIBC treated with chemotherapy + pembrolizumab within the LCCC1520 trial. A high number of analyses was done, including tumor DNA/RNA sequencing, plasma and PBMC profiling, (neo)antigen analysis and a comparison to the ABACUS (anti-PDL1) cohort and an internal chemo-treated cohort. The

authors conclude that “plasma IL-9, tumor IL-8, and tumor stroma-rich subtype represent potential biomarkers of response to NAC-ICI”. These conclusions in my view have a high likelihood of being spurious as many comparisons were done and the cohort was relatively small. Overall, no coherent biological insight or clinically useful conclusion was gained from this study.

Specific comments:

- The cohort was small and results are further clouded by the fact that tumors, (particularly cT2, which was the majority of tumors), could have been removed by TURB alone. As pT1 was included in the path response category, the likelihood of having path response not caused by systemic therapy is even higher.

- It is very difficult in this type of small cohort to know if the results are better than expected with chemotherapy alone, as the clinical results are close to those expected for chemotherapy, especially given the cohort composition in terms of stage. As a result, it is challenging, or rather impossible, to discern any effects of the addition of immunotherapy.

- Some of the main results were from PBMC and peripheral blood analyses. Both systemic therapy modalities used here will have direct effects on the peripheral blood compartment, independent of tumor response. This is particularly true for chemotherapy, which has profound effects on peripheral blood cells (myelosuppression) and on many organs, which could produce cell death-related cytokines. Did any of the results match with therapy effects in the tumor?

- IL-8 is an immune-suppressant cytokine. Whereas I am willing to believe that chemotherapy could compensate for some of the negative IL-8 immune-related effects, I find it hard to understand how these tumors could actually do better than IL-8-low tumors. This conclusion would thus need to be better substantiated. Furthermore, the response in IL-8-high tumors in ABACUS is 36% (15/42), which is not different (or even slightly better) from the response rate in the overall cohort.

- PR and CR are not well-defined in fig 1. Did CR include CIS? Is PR any NMIBC (<pT2 but not CR)? This is mentioned in the text but not in the figure. The same is true for the ABACUS response in fig 6; in this case the response categories are also not mentioned in the text. Importantly, the prim endpoint in ABACUS was pCR (pT0 + CIS) so different response categories may have been used in this figure.

Reviewer #3 (Remarks to the Author): with expertise in immuno-genomics, genitourinary cancers

This is an interesting and well performed analysis of genomic tissue data and blood cytokine and flow cytometry analysis of muscle invasive bladder cancer patients receiving anti-PD1 + gemcitabine. The authors report several interesting changes in tumor mutational burden, blood cytokines and cells after therapy. The main findings being that tumor IL8, plasma IL9 and the tumor being from the stroma rich subtype all predict outcomes to chemo/immunotherapy treatment. Some of the findings are negative, like changes in predicted epitopes after therapy, but I think this is very important to report because other similarly sized studies claim the opposite, and there is no good data to know yet if selection of non-immunogenic tumor clones actually occurs in humans after PDL1 blockade.

Overall the longitudinal samples collected are a very interesting set of samples, the analysis done on the mutational profiles, immune response and correlates with therapy are done well, and the findings might lead to further correlative analysis in future trials. My comments are mostly related to interpretation of some data, and requests to show more of the primary data, none of which alter the main conclusions of the authors.

Fig 1 - Very minor, but fig 1 C and D are switched in text and on the figure. Just throughout the paper there are a few of these. I think text for 6B-C is referring to D, then D is to something different. All very minor, just needs a check.

Fig 2 -The comment below related to Fig 2 is my most major issue that I think needs a lot more clarification and analysis.

“We first analyzed changes in TMB, one of the better pan-cancer predictive biomarkers of response to ICI. TMB decreased between patient-matched pre-treatment and post-treatment tumors (Fig. 2A), suggesting the possibility that immune editing–tumor evolution occurred.”

It might be ok to just drop, but I think the authors are capable of some interesting analysis on these samples to really work out if there is a decline in TMB after therapy, or if its just artifact. Several other studies claim this with similar problems to what I highlight below.

There are alternative explanations for this observation, specifically a reduced ability to

detect mutations with less tumor material after therapy, especially with the switch to more of a stromal subtype (from fig 2F)

i) Does the purity of the tissue post therapy correlate with the decline in tumor cellularity. Presumably the tumor shrank for most of the non-CR patients and it was harder to detect mutations.

ii) same for the change to stromal subtype. My guess is that in these tumors there are far less tumor cells, so probably harder to detect mutations. Are the tumors that became much more stromal having less mutations detected.

iii) Is there a correlation with the size of the tumor material recovered and the number of mutations detected. If small tumors have small numbers of mutations, I think the authors have to consider this is a reduced detection rather than an actual reduction in TMB.

iv) Just generally, its hard to reconcile a decline of probably hundreds of mutations (10 -> 5 TMB) being lost when the predicted epitopes from later figures don't seem to change much. I think some space for discussion here if the authors think their data supports.

v) authors say there is no overall change in 'tumor cellularity', but there are some patients with very dramatic drops. Are these the patients that also have a decline in TMB? What is the correlation between tumor cellularity changes and TMB changes?

Fig 3 – This is a good analysis, but there needs to be figures showing FACS plots of the major populations of immune cells called significant either in supplement or main figure. There needs to be quantification of each of these significant population by patient, showing total percent in blood for each patient. Some of these populations are very rare and of unknown significance in most human blood (e.g PD1+CD45RA+CD4/CD8), so I need to see the FACS plots to be convinced these are real changes.

Do the changes in each cell type correlate together. i.e if a patient has a large increase in CD4+CD45RA-PD1+, do they also have a similar change in CD8+CD45RA-PD1-

Minor, but does the size of the dot in the Fig3 plots just related to the y-axis? Might not need to graph a 3rd variable if its just the same thing, and could be easier to just see if there is less information

If possible, please include the absolute concentration of the chemokines somewhere. These are being increasingly used by others, and good to see what range is being detected by everyone.

Fig 4. Analysis for this is very nice, and I think these authors are doing this as well as any other group by considering self epitopes, structural changes and SNVs. I think the lack of correlation with predicted immunogenic antigen changes probably makes sense, as in many other papers, very few predicted epitopes actually generate a response (e.g see Cauchy et al., Nature, 2020;

This lines up with my comments for Fig 1. Can the authors just explain how there might be a drop in 50% of the total mutations in a tumor if only ~100 are predicted to be immunogenic, and there doesn't seem to be much selection against these epitopes.

Fig 5 – No major issues here. Just interested if the PDL1 expression correlate with the IFN γ signature? Or CD8 infiltration with any signature. Not essential, but if authors think its worth the extra analysis I think a lot of these things just go together and people report them separately.

More discussion of Figure 5G-I would be good. Why do females have such worse survival among non-responders? Why might iDCs and macrophages make for a worse response/survival?

Very minor, but perhaps make the dots in fig 5 similar colors to the groups they belong to. I can tell which dot belongs to which box plot, but they are very close to the neighboring group and sort of mixed in with the neighbors which made it hard to look at quickly.

Fig 6 is outside my knowledge to review, and my only comment was this correlation with response to pre-treatment IL8 is important to discuss because of opposite findings in other tumor types. Authors review this well in the discussion and have provided thoughtful reasoning for the differences.

Reviewer #4 (Remarks to the Author): with expertise in immuno-genomics

In this study, the authors present their findings on the correlation between immune features and response to neoadjuvant chemo-immunotherapy for muscle-invasive bladder cancer. Clinical response was associated with an increase in plasma IL-9 from pre-treatment to pre-cycle 2 of therapy and higher levels of pre-treatment tumor PD-L1 and TIGIT RNA expression. IL-8 signature and Stroma-rich subtype were also associated with improved response rates to NAC-ICI compared to neoadjuvant ICI.

While there are some interesting points, in general I believe there could be significant improvement on how these findings are presented and explained in order to highlight the key results and what this means for future therapy.

Authors discussed their observation of tumor subtype switching and references a separate study (Seiler et al) to suggest that subtype switching could improve patient outcomes. Could the authors elaborate further on how this connection was made? Do subtypes of these post-treatment tumor samples now match the characteristics of pretreatment tumor samples from patients with complete response?

Can the authors clarify what they mean with respect to the statement "...suggesting that if the tumors in patients with recurrent/metastatic disease maintain the effects seen in post-NAC-ICI tumors, then NAC-ICI could reduce ICI efficacy to treat recurrent/metastatic disease". What are the authors trying to suggest here for patient treatment?

In Figure 2A, there seem to be two patients who had an increase in their TMB from pre to post treatment, have the authors explored further into those cases? Additionally, the authors made a rather surprising observation where patients with partial response had a lower TMB compared to those with no response and hypothesized that this could be due to the occurrence of an adverse treatment-related event that required cessation of chemotherapy. Could the authors validate this hypothesis by changing Figure 5B to only include non-responder data points?

The authors present a great amount of analysis results, but don't always elaborate on the implications of the findings. Could the authors elaborate further on the following observations made? Are they expected based on previous literature?

“Finally, we assessed changes in inferred intratumoral BCR and TCR diversity metrics and observed decreased IGK entropy from pre-to post-treatment”

“Finally, we assessed changes in T and B cell receptor diversity from pre-to post-treatment and found no significant changes”

“Overall, pre-treatment TMB was not significantly correlated with Ayers_IFNG signature”

For analysis on tumor antigens burden and its correlation with tumor response, can the authors clarify how they were generated (including specific filters used)? Currently the method section simply states that it's based on the default settings of LENS. Furthermore, it seems unclear what results, from the tumor antigen analysis, the authors are trying to highlight. The authors started off by stating “the number of predicted antigens per patient did not track closely with response” and then continued to characterize the tumor antigens identified, including shared antigen pool. Are these shared antigens only among certain patient groups (by response)? Do the predicted antigens that passed through a more stringent criteria correlate better with patient response?

Figure 6B does not seem to match the authors statements regarding: “Patients with high pre-treatment IL-8 signature had a higher response rate to chemo- immunotherapy , while patients with low pre-treatment IL-8 signature had similar response rates with NAC-ICI and ICI alone”. In high IL-8 signature patients, neoadjuvant ICI seems to perform better than NAC-ICC?

Minor comments:

It would be great if the authors could define all acronyms clearly, including those for immune gene signatures (categorizing them with brief explanations would help with interpretation of results).

Figures and figure captions can be significantly improved (description of axis, sample size, legends etc.)

Figure 2D shows NA category for certain mutations, what do those entail?

Figure 4A columns seem to be sorted based on self-antigen counts, have the authors tried using clinical response groups to arrange the columns?

REVIEWER COMMENTS

Reviewer #1 (Remarks to the Author): with expertise in bladder cancer (therapy, omics)

Comments and questions to the authors:

This is a relevant and comprehensive molecular investigation patients treated with neoadjuvant chemo- and immunotherapy. The data is clearly presented and the analysis are in line with clinically relevant questions and the conclusions are supported by the findings.

1. Do the authors have information on purity of tumor samples? If not, the reviewer suggests to add this information and compare to present data (e.g. molecular subtype, TMB, immune gene signatures).

We appreciate the suggestion from Reviewer 1 to include tumor purity data. From pre-and post-treatment H&E slides, we had 5 remaining paired pre-and post-treatment slides available for analysis. A pathologist [SW] assessed tumor purity for each of these 10 slides. We then compared tumor purity by molecular subtype, TMB, and 3 immune gene signatures: Bindea_CD8_T_cells, Ayers_IFNG, and IL-8 signature (Supp. Fig. 1E-I).

We added the following description of tumor purity analysis to the Methods section: **“Tumor purity analysis: Tumor purity was assessed for 5 tumors with matched pre-and post-treatment samples. An expert pathologist [SW] estimated tumor counts (papillary and invasive) as a percentage of total cells on formalin-fixed paraffin-embedded slides stained with Hematoxylin & Eosin.”** We added the following text to Results: (Lines 191-196) **“The decrease in TMB was not correlated with changes in tumor purity estimated from pathology (Supp. Fig. 2A) or tumor cellularity calculated by Sequenza (Supp. Fig. 2B-C). Furthermore, the decrease in TMB does not appear to be confounded by reduced detection of mutations in small tumors, because the number of mutations in paired pre-and post-treatment NR and PR tumors is not**

significantly different at post-treatment (Supp. Fig. 2D), and the difference between median read depth in pre-treatment (149.5) and post-treatment (136) tumors at shared mutational loci is small (Supp. Fig. 2E).”

2. The authors state that subtype switching was not associated with tumor cellularity but use gene expression data to underline this statement. The reviewer suggests to support this statement by data on purity of samples or IHC.

We would like to thank Reviewer 1 for making the important point that additional evidence is needed to support our claim that subtype switching is independent of changes in tumor cellularity. We now support the statement with the tumor purity data, comparing changes in tumor purity in patients with or without subtype switching (Supp. Fig. 1E).

3. The authors performed flow cytometry and included approx. 43 antibodies. Were samples aliquoted and analyses separate for T-, B- and myeloid cells? If so, then this should more clearly

be described in the material and method section. In addition, how did the authors standardize flow cytometry values for analyses between different aliquotes (e.g. for Figure 3A)?

We appreciate the request from Reviewer 1 to include information on cell aliquotes in the flow cytometry methods section. We added the following to Methods: (Line 148) **“Samples were aliquoted and analyses performed separately for T, B, and myeloid cells.”**

4. The authors further analysed features that were associated with survival in non-responders and identified that male patients survived longer. Which other parameters were investigated? This information could e.g. be give in an extended table.

We agree with Reviewer 1 that this information is important to include to provide context for the claim regarding survival in male patients. In response, we included a supplementary table (Supp. Table 5) comparing the effects of other features on survival.

Supplementary Table 5. Clinical feature associations with survival.

Feature	Hazard ratio	p-value
Received prior intravesical therapy	0.3	0.2
Former smoker	0.6	0.5
Current smoker	1.2	0.8
Years smoked	1.0	0.5
ECOG-PS = 1	0.3	0.2

5. Do the authors see other options, why tumors lost TP53 and KDM6A under treatment? They suggest immune editing, could treatment induced clonal selection play a role?

We appreciate the suggestion from Reviewer 1 that we consider the role clonal selection could play in tumor loss of TP53 and KDM6A. In response, we added the following to the Discussion section: (Lines 317-321) **“However, 6 of 10 TP53-mutant tumors lost their TP53 mutations between pre-and post-treatment, and 4/5 KDM6A-mutant tumors lost their KDM6A mutations between pre-and post-treatment, suggesting that immune editing could have occurred. Alternatively, treatment-induced clonal selection could have occurred: TURBT could be a bottleneck event, where there is only clonal outgrowth from a small population of a heterogeneous original tumor.”**

6. The authors performed elastic net modeling to evaluate features for their associations with response. Did the authors try to build a model based on all of these parameters to predict response (e.g. random forest, GLM or similar)? Internal crossvalidation (e.g. leave one out) would allow to get information on the performance of such a model. The accuracy of such a model, trained on a comprehensive panel of biomarkers might be of interest for researcher and help to guide biobanking for further clinical trials.

We agree with Reviewer 1 that a model trained on a comprehensive panel of biomarkers could be important for future research, and although the LCCC1520 trial is relatively small, a model with internal cross-validation provides some insight into which biomarkers could predict response in future trials. While we cannot draw strong conclusions from a model that is not validated on an independent data set, we appreciate the value in including such an analysis. For our elastic net GLM model, we presented the coefficients and confidence intervals of the model from internal 10-fold cross-validation, and then we used the mean values from the cross-validated models to predict response (Fig. 5I), demonstrating that the variables and coefficients selected by the model are associated with response in LCCC1520.

7. The conclusions of the abstract and the summary of the main manuscripts are quite different. Both seem to be appropriate but the reader gets different information when reading only one or the other. The reviewer suggests to modify either one or the other.

We appreciate this comment from Reviewer 1 and apologize for the confusion of the discordant Abstract and Summary conclusions. We modified the Abstract conclusions to better match the Summary conclusions: (Lines 41-45) **“Plasma IL-9 represents a potential predictive biomarker of response to NAC-ICI. Tumor IL-8 signature and stroma-rich subtype represent potential predictive biomarkers of response benefit of NAC-ICI over neoadjuvant ICI. Future efforts must include further independent biomarker discovery and validation, including within the context of prospective clinical trials, ultimately to improve the selection of patients for ICI-related treatments.”**

Minor:

- Results, clinical features of NAC-ICI, 2nd line: (Rose et al), other refs are given in numbers in superscript.

We would like to thank Reviewer 1 and apologize for this error. We have fixed the error by placing the reference number in superscript.

- The CCR4-Scar-like subtype is not described in the here given reference.

We would like to thank Reviewer 1 and apologize for this improper citation. We have fixed the citation to cite the correct manuscript.

Reviewer #2 (Remarks to the Author): with expertise in bladder cancer, (immuno)therapy

Specific comments:

- The cohort was small and results are further clouded by the fact that tumors, (particularly cT2, which was the majority of tumors), could have been removed by TURB alone. As pT1 was included in the path response category, the likelihood of having path response not caused by systemic therapy is even higher.

We agree with Reviewer 2 that it is important to acknowledge that TURB alone can remove tumors and result in pathologic downstaging at cystectomy. This is an important consideration to contextualize our results. However, we also acknowledge that regardless of the main drivers of downstaging, patients with pathologic downstaging had significantly longer survival. We added further commentary on this subject in the Limitations section of the Discussion: (Lines 388-391) **“While analysis of pathologic response is confounded by the fact that TURBT removes much or all of the tumor, our data adds evidence that there are immune-related predictors of response to TURBT plus NAC-ICI, suggesting that not only TURBT but also the systemic treatment associated with longer survival.”**

-It is very difficult in this type of small cohort to know if the results are better than expected with chemotherapy alone, as the clinical results are close to those expected for chemotherapy, especially given the cohort composition in terms of stage. As a result, it is challenging, or rather impossible, to discern any effects of the addition of immunotherapy.

We agree with Reviewer 2 that it is impossible to discern the benefit of immunotherapy definitively in a single-arm trial of chemoimmunotherapy. However, this manuscript does provide evidence that immunotherapy could be inducing biological effects (e.g. decreased %PD-1+ of CD8 T cells) that are consistent with improved tumor response. To clarify this point, we added the following text to the Discussion: (Lines 363-366) **“While it is impossible to discern definitively the benefit of the immunotherapy in NAC-ICI, this manuscript provides evidence that immunotherapy could have effects (e.g. decreased % PD-1+ CD8 T cells), and these effects are consistent with improved tumor response.”** Moreover, we compare predictive features between our NAC-ICI trial and the ABACUS neoadjuvant ICI trial and find that there are features that are predictive of response to NAC-ICI but not ICI alone (tumor IL8 signature and Consensus stroma-rich subtype). Therefore, there do appear to be distinct pre-treatment features that associate with response and thus suggest differences in underlying biology that affect response to NAC-ICI and neoadjuvant ICI.

- Some of the main results were from PBMC and peripheral blood analyses. Both systemic therapy modalities used here will have direct effects on the peripheral blood compartment,

independent of tumor response. This is particularly true for chemotherapy, which has profound effects on peripheral blood cells (myelosuppression) and on many organs, which could produce cell death-related cytokines. Did any of the results match with therapy effects in the tumor?

We appreciate this comment from Reviewer 2 on the systemic effects of chemotherapy. Being able to track changes in PBMCs and peripheral blood that correlate with tumor changes is an important area of ongoing research toward adaptive therapy. The changes we observed in IL-9 plasma concentrations are associated with tumor response. We are unable to assess how intratumoral IL9 levels change with therapy because responders (at least those with CR) unfortunately have no tumor left post-treatment to examine.

- IL-8 is an immune-suppressant cytokine. Whereas I am willing to believe that chemotherapy could compensate for some of the negative IL-8 immune-related effects, I find it hard to understand how these tumors could actually do better than IL-8-low tumors. This conclusion would thus need to be better substantiated. Furthermore, the response in IL-8-high tumors in ABACUS is 36% (15/42), which is not different (or even slightly better) from the response rate in the overall cohort.

We agree with Reviewer 2's previous comment that it is difficult to compare trial response rates head-to-head because of cohort and drug differences. We also agree with this comment from Reviewer 2 that the effect of IL-8 we present here is counterintuitive, since IL-8 is an immune-suppressant cytokine. To address this comment, we postulate in the Discussion that if IL-8 is driving tumor resistance to immunotherapy, then adding chemotherapy to immunotherapy could provide differential benefit: (Lines 353-356) **“Chemotherapy can decrease specific immunosuppressive myeloid populations, including MDSCs⁵¹, suggesting that the response benefit in chemo-immunotherapy in IL-8 signature-high tumors could be due to the effects of chemotherapy on cells that inhibit ICI. If IL-8 is driving tumor immune resistance, then adding chemotherapy to immunotherapy could provide differential benefit.”**

-PR and CR are not well-defined in fig 1. Did CR include CIS? Is PR any NMIBC (<pT2 but not CR)? This is mentioned in the text but not in the figure. The same is true for the ABACUS response in fig 6; in this case the response categories are also not mentioned in the text. Importantly, the prim endpoint in ABACUS was pCR (pT0 + CIS) so different response categories may have been used in this figure.

We would like to thank Reviewer 2 for this feedback and apologize for the confusion regarding response category definitions. To clarify, we defined response categories more clearly in the Methods section: (Lines 100-101) **“Response criteria: We defined response as <pT2N0 at cystectomy. We divided response into partial response (>pT0 at cystectomy) and complete response (pT0 at cystectomy).”**

We also reiterated these definitions in Figure 1.

Reviewer #3 (Remarks to the Author): with expertise in immuno-genomics, genitourinary cancers

Fig 1 - Very minor, but fig 1 C and D are switched in text and on the figure. Just throughout the

paper there are a few of these. I think text for 6B-C is referring to D, then D is to something different. All very minor, just needs a check.

We thank Reviewer 3 for pointing out this error and apologize for the confusion. We updated the figure numbering for Figures 1 and 6 to match in the text and in the figures. We also added the following text to describe Figure 6B: (Lines 289-291) **“Patients with high pre-treatment IL-8 signature had a higher response rate to NAC-ICI, while patients with low pre-treatment IL-8 signature had similar response rates with NAC-ICI or ICI alone (Fig. 6B-D).”**

Fig 2 -The comment below related to Fig 2 is my most major issue that I think needs a lot more clarification and analysis.

“We first analyzed changes in TMB, one of the better pan-cancer predictive biomarkers of response to ICI. TMB decreased between patient-matched pre-treatment and post-treatment tumors (Fig. 2A), suggesting the possibility that immune editing–tumor evolution occurred.” It might be ok to just drop, but I think the authors are capable of some interesting analysis on these samples to really work out if there is a decline in TMB after therapy, or if its just artifact. Several other studies claim this with similar problems to what I highlight below. There are alternative explanations for this observation, specifically a reduced ability to detect mutations with less tumor material after therapy, especially with the switch to more of a stromal subtype (from fig 2F)

i) Does the purity of the tissue post therapy correlate with the decline in tumor cellularity. Presumably the tumor shrank for most of the non-CR patients and it was harder to detect mutations.

We appreciate this question and agree that technical differences in tumor cell abundance could have affected the mutation detection. To address this, we first analyzed the correlation of tumor purity with tumor cellularity post therapy (Supp. Fig. 2A) and found no strong correlation between the two metrics.

Next, we compared tumor sample read depth at loci of mutations shared between pre-and post-treatment samples. Post-treatment NR samples do not have significantly higher mutation count than PR samples (Supp. Fig. 2D). While pre-treatment samples had significantly higher read depth, the absolute difference in median read depth between pre-treatment (149.5) and post-treatment (136) was small and could not fully explain the reduction in mutational detection (Supp. Fig. 2E).

li) same for the change to stromal subtype. My guess is that in these tumors there are far less tumor cells, so probably harder to detect mutations. Are the tumors that became much more stromal having less mutations detected.

From pre-and post-treatment H&E slides, we had 5 remaining paired pre-and post-treatment slides available for analysis. A pathologist [SW] assessed tumor purity for each of these 10 slides. In the 3 tumors out of the 5 that changed to stromal subtype, only one had reduced tumor purity, suggesting that the change to stromal subtype cannot be fully explained by a reduction in tumor purity (Supp. Fig. 1E).

We also assessed mutation count post-treatment and found that the tumors that switched to stroma-rich subtype did not have fewer mutations detected (Supp. Fig. 1J). Finally, we compared changes in TMB and mutation count in the samples tested by pathology for tumor purity. We found that post-treatment mutation count (Supp. Fig. 1K) and TMB (Supp. Fig. 1L) were not lower in the tumors that switched to stroma-rich subtype that were tested by pathology for tumor purity.

iii) Is there a correlation with the size of the tumor material recovered and the number of mutations detected. If small tumors have small numbers of mutations, I think the authors have to consider this is a reduced detection rather than an actual reduction in TMB.

We agree with Reviewer 3 that size of tumor material recovered is an important confounding variable to consider that could affect TMB calculation. We found no significant difference between post-treatment mutation count in PR and NR tumors, suggesting that mutation count in post-treatment tumors is independent of tumor size (Supp. Fig. 2D).

iv) Just generally, its hard to reconcile a decline of probably hundreds of mutations (10 -> 5 TMB) being lost when the predicted epitopes from later figures don't seem to change much. I think some space for discussion here if the authors think their data supports.

We agree with Reviewer 3 that this is an important point of discussion. One possibility is that the TURBT represents a bottleneck event, where there is only clonal regrowth from a small population derived from a heterogeneous original tumor. We included this hypothesis in the Discussion: (Lines 319-321) **“Alternatively, treatment-induced clonal selection could have occurred: TURBT could be a bottleneck event, where there is only clonal outgrowth from a small population of a heterogeneous original tumor.”**

v) authors say there is no overall change in ‘tumor cellularity’, but there are some patients with very dramatic drops. Are these the patients that also have a decline in TMB? What is the correlation between tumor cellularity changes and TMB changes?

We appreciate that Reviewer 3 brings up the fact that tumor cellularity changes vary dramatically between patients. To address this point, we assessed the correlation between changes in TMB and changes in tumor cellularity (Supp. Fig. 2B) and found no strong

correlation.

Fig 3 – This is a good analysis, but there needs to be figures showing FACS plots of the major populations of immune cells called significant either in supplement or main figure. There needs to be quantification of each of these significant population by patient, showing total percent in blood for each patient. Some of these populations are very rare and of unknown significance in most human blood (e.g PD1+CD45RA+CD4/CD8), so I need to see the FACS plots to be convinced these are real changes.

We would like to thank Reviewer 3 for requesting further documentation of the flow cytometry data, with FACS plots and population quantification, because this adds further evidence that these populations are abundant enough to be biologically relevant. To address these requests, we added example FACS plots for all the flow cytometry populations (Supp. Fig. 3A-C), as well as quantification of percent of live cells in the blood for each patient (Supp. Fig. 3D).

Do the changes in each cell type correlate together. i.e if a patient has a large increase in CD4+CD45RA-PD1+, do they also have a similar change in CD8+CD45RA-PD1-

We appreciate that Reviewer 3 raises the point that some of the changes in immune populations could be correlated with each other. To address this point, we assessed the correlations of the 10 immune populations with significant changes and display this information in a heatmap (Supp. Fig. 3E).

Minor, but does the size of the dot in the Fig3 plots just related to the y-axis? Might not need to graph a 3rd variable if its just the same thing, and could be easier to just see if there is less information

We agree with Reviewer 3 that differences in dot sizes in the Figure 3 (and Figure 2) plots are unnecessary and detract from the clarity of the figure. In response, we removed the variability in dot sizes from these plots.

If possible, please include the absolute concentration of the chemokines somewhere. These are being increasingly used by others, and good to see what range is being detected by everyone.

We would like to thank Reviewer 3 for requesting we include this data for future comparison. We included the absolute concentrations of the chemokines in a supplementary table (Supplementary Table 6 [attached as spreadsheet: "1520plasma.csv"]).

Fig 4. Analysis for this is very nice, and I think these authors are doing this as well as any other group by considering self epitopes, structural changes and SNVs. I think the lack of correlation with predicted immunogenic antigen changes probably makes sense, as in many other papers, very few predicted epitopes actually generate a response (e.g see Cauchy et al., Nature, 2020; This lines up with my comments for Fig 1. Can the authors just explain how there might be a drop in 50% of the total mutations in a tumor if only ~100 are predicted to be immunogenic, and there doesn't seem to be much selection against these epitopes.

We thank the reviewer for this comment and agree with the alignment with the Cauchy et al data. As above, we agree with Reviewer 3 that this is an important point of discussion. One possibility is that the TURBT represents a bottleneck event, where there is only clonal regrowth from a small population derived from a heterogeneous original tumor. We included this hypothesis in the Discussion: (Lines 319-321) "**Alternatively, treatment-induced clonal**

selection could have occurred: TURBT could be a bottleneck event, where there is only clonal outgrowth from a small population of a heterogeneous original tumor.”

Fig 5 – No major issues here. Just interested if the PDL1 expression correlate with the IFNg signature? Or CD8 infiltration with any signature. Not essential, but if authors think its worth the extra analysis I think a lot of these things just go together and people report them separately.

We agree with Reviewer 3 that these correlations are interesting and should be included in the manuscript. We assessed the correlations between PD-L1 and IFNg, and between CD8 infiltration and the other immune gene signatures. We included these correlations as supplementary figures (Supp. Fig. 5B-C). We also described our assessment of these correlations in Results: (Lines 261-264) **“PD-L1 expression and Ayers_IFNG signature were significantly correlated with each other (Supp. Fig. 5C), and CD8 infiltration (Bindea_CD8_T_cells) was significantly associated with many of the other immune gene signatures (Supp. Fig. 5D).”**

More discussion of Figure 5G-I would be good. Why do females have such worse survival among non-responders? Why might iDCs and macrophages make for a worse response/survival?

We appreciate Reviewer 3’s request for us to discuss further why females might have worse survival among non-responders. In response, we added the following to the Discussion: (Lines 367-372) **“Several other potential biomarkers associated with outcome. Females had worse survival among non-responders. While there are too few females in the trial to assess factors influencing sex-specific differential survival by response, these data suggest that a potential biomarker of response might differ by sex. Additionally, iDC and macrophages were associated with worse outcomes in our multivariable model. iDCs⁵² and tumor-associated macrophages⁵³ both have potential roles in attenuating tumor responses to ICB.”**

Very minor, but perhaps make the dots in fig 5 similar colors to the groups they belong to. I can tell which dot belongs to which box plot, but they are very close to the neighboring group and sort of mixed in with the neighbors which made it hard to look at quickly.

We agree with Reviewer 3 that the coloring the dots by group improves the legibility of Figure 5. We revised Figure 5 so that the dot colors match the box plot colors and made the widths of the dot ranges smaller so it is more clear which dots correspond to each box plot.

Fig 6 is outside my knowledge to review, and my only comment was this correlation with response to pre-treatment IL8 is important to discuss because of opposite findings in other tumor types. Authors review this well in the discussion and have provided thoughtful reasoning for the differences.

Reviewer #4 (Remarks to the Author): with expertise in immuno-genomics

Authors discussed their observation of tumor subtype switching and references a separate study (Seiler et al) to suggest that subtype switching could improve patient outcomes. Could the authors elaborate further on how this connection was made? Do subtypes of these post-treatment tumor samples now match the characteristics of pretreatment tumor samples from patients with complete response?

We agree with Reviewer 4 that further elaboration is necessary to explain how subtype switching could improve patient outcomes. In response to this comment, we included the following in the Discussion: (Lines 309-312) **“Because many of the tumors in LCCC1520 switched to a Stroma-rich subtype similar to the CCR4-Scar-like phenotype assessed by Seiler et al, we hypothesize that subtype switching on neoadjuvant chemo-immunotherapy could improve patient outcomes post-therapy.”**

Can the authors clarify what they mean with respect to the statement “...suggesting that if the tumors in patients with recurrent/metastatic disease maintain the effects seen in post-NAC-ICI tumors, then NAC-ICI could reduce ICI efficacy to treat recurrent/metastatic disease”. What are the authors trying to suggest here for patient treatment?

We thank Reviewer 4 for noting that this phrase is confusing, and we apologize for the lack of clarity. We removed the statement from the Results section and added the following statement to a paragraph in the Discussion on Findings of Unknown Significance: (Lines 374-376) **“High FTBRS signature⁴ and low TCGA_IFNG⁵⁵ have been associated with worse response to ICI, suggesting that the post-treatment tumors might respond worse to any future ICI treatment than if they had not been treated with NAC-ICI.”**

In Figure 2A, there seem to be two patients who had an increase in their TMB from pre to post treatment, have the authors explored further into those cases?

We appreciate this request from Reviewer 4 to further analyze data from the two patients who had an increase in TMB from pre to post treatment. In response to this request, we included additional supplementary figures (Supp. Fig. 2G-I) presenting associations of key potential predicted biomarkers in those two patients versus in the rest of the cohort of non-pCR patients.

Additionally, the authors made a rather surprising observation where patients with partial response had a lower TMB compared to those with no response and hypothesized that this could be due to the occurrence of an adverse treatment-related event that required cessation of chemotherapy. Could the authors validate this hypothesis by changing Figure 5B to only include non-responder data points?

We agree with Reviewer 4 that the association between TMB and adverse treatment-related events is surprising, interesting, and warrants further investigation. Per request, we replotted Figure 5B to only include non-responder data points and did not find a significant association between adverse events and TMB. We included this figure in supplement (Supp. Fig. 5A).

The authors present a great amount of analysis results, but don't always elaborate on the implications of the findings. Could the authors elaborate further on the following observations

made? Are they expected based on previous literature?

“Finally, we assessed changes in inferred intratumoral BCR and TCR diversity metrics and observed decreased IGK entropy from pre-to post-treatment”

“Finally, we assessed changes in T and B cell receptor diversity from pre-to post-treatment and found no significant changes”

“Overall, pre-treatment TMB was not significantly correlated with Ayers_IFNG signature”

In response to these 3 observations, we included an additional paragraph in the Discussion on Findings of Unknown Significance: (Lines 373-385) **“We also made several findings of unknown significance. First, we observed increases in FTBRS signature and decreases in TCGA_IFNG signature across treatment. High FTBRS signature⁴ and low TCGA_IFNG⁵⁵ have been associated with worse response to ICI, suggesting that the post-treatment tumors might respond worse to any future ICI treatment than if they had not been treated with NAC-ICI. Second, we observed decreased IGK entropy from pre-to post-treatment but no significant changes in BCR or TCR diversity. While CDR3 sequences from IGK are abundant (54% of the B cell population), no studies to our knowledge have documented that decreases in the entropy of IGK are associated with NAC-ICI therapy. Third, we observed that pre-treatment TMB was not significantly correlated with Ayers_IFNG signature, consistent with prior literature demonstrating poor correlation between mutational burden and cytolytic signatures^{56,57}. While both TMB and Ayers_IFNG signature have been proposed as potential predictive biomarkers of ICI response^{18,58}, we report differential associations of the two markers with NAC-ICI response. Further clinical correlative analysis from future clinical trials will need to be conducted to contextualize and to evaluate the generalizability of these findings.”**

For analysis on tumor antigens burden and its correlation with tumor response, can the authors clarify how they were generated (including specific filters used)? Currently the method section simply states that it's based on the default settings of LENS. Furthermore, it seems unclear what results, from the tumor antigen analysis, the authors are trying to highlight. The authors started off by stating “the number of predicted antigens per patient did not track closely with response” and then continued to characterize the tumor antigens identified, including shared antigen pool. Are these shared antigens only among certain patient groups (by response)? Do the predicted antigens that passed through a more stringent criteria correlate better with patient response?

We appreciate the request from Reviewer 4 to include better detailed methods for the antigen selection criteria in LENS, since transparency and adaptability of the LENS platform built by our group are two of its strengths. To address Reviewer 4's comments here, we added further description in a Supplementary Methods section on the filters and packages used for LENS (Lines 697-749). In the Results paragraph, we provided context for the analysis by mentioning the reasoning behind analyzing shared antigens and mutation etiologies: (Lines 237-238) **“We further assessed the number of shared predicted antigens because these could be attractive targets for development of biomarkers or vaccines that could help multiple patients.”** (Lines 244-247) **“To assess whether certain mutational etiologies are targeted during chemo-immunotherapy response, we analyzed the COSMIC signatures of SNV and indel mutations**

and found a large proportion of the mutational landscape in both the pre-and post-treatment samples is attributable to APOBEC activity (Fig. 4G).”

We also included a supplementary figure (Supp. Fig. 4B) to demonstrate that there is no better response correlation with the more stringent TESLA antigen selection criteria. We included the following statement in the Results section: (Lines 235-237) **“However, the number of predicted antigens per patient did not track closely with response (Supp. Fig. 4B), TMB, Ayers IFNG signature, or CD8 T cell signature (Fig. 4A).”**

Finally, in the Discussion section, we mentioned that we did not find any associations of antigen types or counts with response: (Lines 342-345) **“We analyzed shared antigens because these are attractive targets for development of biomarkers or vaccines that could help multiple patients. We did not find any associations between antigen types or counts and response.”**

Figure 6B does not seem to match the authors statements regarding: “Patients with high pre-treatment IL-8 signature had a higher response rate to chemo- immunotherapy, while patients with low pre-treatment IL-8 signature had similar response rates with NAC-ICI and ICI alone”. In high IL-8 signature patients, neoadjuvant ICI seems to perform better than NAC-ICC?

We thank Reviewer 4 for highlighting an error with the labels for Figure 6B. We fixed the labeling error. For clarity, we also revised the figure labels to read “NAC-ICI” and “Neoadjuvant ICI” instead of “LCCC1520” and “ABACUS”.

Minor comments:

It would be great if the authors could define all acronyms clearly, including those for immune gene signatures (categorizing them with brief explanations would help with interpretation of results).

We thank Reviewer 4 for pointing out the need for acronym definitions throughout. We defined all acronyms at their first instances and included brief explanations of all the immune gene signatures mentioned in the text (e.g. Lines 204-209) **“From pre-to post-treatment, signatures for Innate anti-PD-1 Resistance non-response (IPRES_NonResponder), fibroblast TGF-beta response (FTBRS), and natural killer cells (Bindea_NK_cells) increased, while signatures for immature dendritic cells (Bindea_iDC), T helper 2 cells (Bindea_Th2_cells), decreased**

epithelial-mesenchymal transition (EMT_DOWN), B cells (BCell_60gene), and IFN γ (TCGA_IFNG) decreased (FDR-corrected $p < 0.05$).

Figures and figure captions can be significantly improved (description of axis, sample size, legends etc.)

We appreciate Reviewer 4's request to improve figures and figure captions. We revised the figure axes to be in bold and with consistent size, and we revised the figure legends to contain consistent labels. We revised the figure captions to include sample size and descriptions of the axes.

Figure 2D shows NA category for certain mutations, what do those entail?

We thank Reviewer 4 for noting that we do not define clearly what the NA category entails in Figure 2D. We updated Figure 2D to have the correct category labels for all the mutations gained or lost, including those mutations in multi-hit tumors.

Figure 4A columns seem to be sorted based on self-antigen counts, have the authors tried using clinical response groups to arrange the columns?

We agree with Reviewer 4 that arranging the self-antigen counts by clinical response groups is another intuitive way to arrange the figure panel. Since there is not a significant association between clinical response and self-antigen counts, we included the figure panel arranged by clinical response group as a supplementary figure (Supp. Fig. 4A).

REVIEWERS' COMMENTS

Reviewer #1 (Remarks to the Author):

The authors addressed the comments of the reviewer and modified the manuscript accordingly.

Reviewer #2 (Remarks to the Author):

My concern remains that a high number of analyses are done on a small cohort of patients and there is a high likelihood of findings being spurious.

The only way to reach a bit more certainty would be to validate on a separate cohort (several chemo-immunotherapy studies have been published).

Reviewer #3 (Remarks to the Author):

In my initial review I had asked the authors for several pieces of additional analysis to rule out various technical issues that might explain some of the observations reported. The authors have addressed all these the best they can from the data and provided reasonable additional discussion around the points raise. Work is really interesting overall and certainly suitable for publication

Reviewer #4 (Remarks to the Author):

My questions have been addressed and I have no further comments.

Reviewer #4 (Remarks on code availability):

The R code within the txt file that the authors provided looks reasonable but I had trouble running the code and generating the corresponding plots. In particular, 'clinical1520.csv' could not be found in the Source_Data folder provided by the authors.

REVIEWERS' COMMENTS

Reviewer #1 (Remarks to the Author):

The authors addressed the comments of the reviewer and modified the manuscript accordingly.

Reviewer #2 (Remarks to the Author):

My concern remains that a high number of analyses are done on a small cohort of patients and there is a high likelihood of findings being spurious.

The only way to reach a bit more certainty would be to validate on a separate cohort (several chemo-immunotherapy studies have been published).

We thank Reviewer #2 for their feedback. Unfortunately, There are no other bladder cancer chemo-immunotherapy studies with publicly available sequencing data, so we are unable to validate on a separate cohort now. We updated the Limitations section of the Discussion to acknowledge this limitation:

Lines 288-291: “While analysis of the correlative data from this trial of neoadjuvant chemo-immunotherapy presents interesting findings, it has several limitations. The trial is small, retrospective, and has limited data for certain analyses, increasing the likelihood of spurious findings. Validation on separate chemo-immunotherapy cohorts in the future will be necessary.”

Reviewer #3 (Remarks to the Author):

In my initial review I had asked the authors for several pieces of additional analysis to rule out various technical issues that might explain some of the observations reported. The authors have addressed all these the best they can from the data and provided reasonable additional discussion around the points raise. Work is really interesting overall and certainly suitable for publication

Reviewer #4 (Remarks on code availability):

The R code within the txt file that the authors provided looks reasonable but I had trouble running the code and generating the corresponding plots. In particular, 'clinical1520.csv' could not be found in the Source_Data folder provided by the authors.

We thank Reviewer #4 for reviewing the code associated with the manuscript. We have now added the 'clinical1520.csv' clinical data file and updated the column references in the code to match the format of the clinical data file.